# NTK-SAP: Improving neural network pruning by aligning training dynamics

**Yite Wang**[1]**, Dawei Li**[1]**, Ruoyu Sun**[2,3*]
[1]University of Illinois Urbana-Champaign, USA
[2]Shenzhen International Center for Industrial and Applied Mathematics,
  Shenzhen Research Institute of Big Data
[3]School of Data Science, The Chinese University of Hong Kong, Shenzhen, China
`{yitew2,dawei2}@illinois.edu, sunruoyu@cuhk.edu.cn`

## Abstract

Pruning neural networks before training has received increasing interest due to its potential to reduce training time and memory. One popular method is to prune the connections based on a certain metric, but it is not entirely clear what metric is the best choice. Recent advances in neural tangent kernel (NTK) theory suggest that the training dynamics of large enough neural networks is closely related to the spectrum of the NTK. Motivated by this finding, we propose to prune the connections that have the least influence on the spectrum of the NTK. This method can help maintain the NTK spectrum, which may help align the training dynamics to that of its dense counterpart. However, one possible issue is that the fixed-weight-NTK corresponding to a given initial point can be very different from the NTK corresponding to later iterates during the training phase. We further propose to sample multiple realizations of random weights to estimate the NTK spectrum. Note that our approach is weight-agnostic, which is different from most existing methods that are weight-dependent. In addition, we use random inputs to compute the fixed-weight-NTK, making our method data-agnostic as well. We name our foresight pruning algorithm Neural Tangent Kernel Spectrum-Aware Pruning (NTK-SAP). Empirically, our method achieves better performance than all baselines on multiple datasets. Our code is available at `https://github.com/YiteWang/NTK-SAP`.

## 1 Introduction

The past decade has witnessed the success of deep neural networks (DNNs) in various applications. Modern DNNs are usually highly over-parameterized, making training and deployment computationally expensive. Network pruning has emerged as a powerful tool for reducing time and memory costs. There are mainly two types of pruning methods: post-hoc pruning (Han et al., 2015; Renda et al., 2020; Molchanov et al., 2019; LeCun et al., 1989; Hassibi & Stork, 1992) and foresight pruning (Lee et al., 2018; Wang et al., 2020; Alizadeh et al., 2022; Tanaka et al., 2020; de Jorge et al., 2020b; Liu & Zenke, 2020). The former methods prune the network after training the large network, while the latter methods prune the network before training. In this paper, we focus on foresight pruning.

SNIP (Lee et al., 2018) is probably the first foresight pruning method for modern neural networks. It prunes those initial weights that have the least impact on the initial loss function. SNIP can be viewed as a special case of saliency score based pruning: prune the connections that have the least "saliency scores", where the saliency score is a certain metric that measures the importance of the connections. For SNIP, the saliency score of a connection is the difference of loss function before and after pruning this connection. GraSP (Wang et al., 2020) and Synflow (Tanaka et al., 2020) used two different saliency scores. One possible issue of these saliency scores is that they are related to the initial few steps of training, and thus may not be good choices for later stages of training. Is there a saliency score that is more directly related to the whole training dynamics?

---

*Corresponding author.

## 1.1 OUR METHODOLOGY

Recently, there are many works on the global optimization of neural networks; see, e.g., Liang et al. (2018b;a; 2019; 2021); Venturi et al. (2018); Safran & Shamir (2017); Li et al. (2022); Ding et al. (2022); Soltanolkotabi et al. (2019); Sun et al. (2020a); Lin et al. (2021a;b); Zhang et al. (2021) and the surveys Sun et al. (2020b); Sun (2020). Among them, one line of research uses neural tangent kernel (NTK) (Jacot et al., 2018) to describe the gradient descent dynamics of DNNs when the network size is large enough. More specifically, for large enough DNNs, the NTK is asymptotically constant during training, and the convergence behavior can be characterized by the spectrum of the NTK. This theory indicates that the spectrum of the NTK might be a reasonable metric for the whole training dynamics instead of just a few initial iterations. It is then natural to consider the following conceptual pruning method: prune the connections that have the least impact on the NTK spectrum.

There are a few questions on implementing this conceptual pruning method.

First, what metric to compute? Computing the whole eigenspectrum of the NTK is too time-consuming. Following the practice in numerical linear algebra and deep learning (Lee et al., 2019a; Xiao et al., 2020), we use the nuclear norm (sum of eigenvalues) as a scalar indicator of the spectrum.

Second, what "NTK" matrix to pick? We call the NTK matrix defined for the given architecture with a random initialization as a "fixed-weight-NTK", and use "analytic NTK" (Jacot et al., 2018) to refer to the asymptotic limit of the fixed-weight-NTK as the network width goes to infinity. The analytic NTK is the one studied in NTK theory (Jacot et al., 2018), and we think its spectrum may serve as a performance indicator of a certain architecture throughout the whole training process.[1] However, computing its nuclear norm is still too time-consuming (either using the analytic form given in Jacot et al. (2018) or handling an ultra-wide network). The nuclear norm of a fixed-weight-NTK is easy to compute, but the fixed-weight-NTK may be quite different from the analytic NTK. To resolve this issue, we notice a less-mentioned fact: the analytic NTK is also the limit of the expectation (over random weights) of fixed-weight-NTK[2], and thus it can be approximated by the expectation of fixed-weight-NTK for a given width. The expectation of fixed-weight-NTK shall be a better approximation of analytic NTK than a single fixed-weight-NTK. Of course, to estimate the expectation, we can use a few "samples" of weight configurations and compute the average of a few fixed-weight-NTKs.

One more possible issue arises: would computing, say, 100 fixed-weight-NTKs take 100 times more computation cost? We use one more computation trick to keep the computation cost low: for each mini-batch of input, we use a fresh sample of weight configuration to compute one fixed-weight-NTK (or, more precisely, its nuclear norm). This will not increase the computation cost compared to computing the fixed-weight-NTK for one weight configuration with 100 mini-batches. We call this "*new-input-new-weight*" (NINW) trick.

We name the proposed foresight pruning algorithm **N**eural **T**angent **K**ernel **S**pectrum-**A**ware **P**runing (**NTK-SAP**). We show that NTK-SAP is competitive on multiple datasets, including CIFAR-10, CIFAR-100, Tiny-ImageNet, and ImageNet. In summary, our contributions are:

- We propose a theory-motivated foresight pruning method named NTK-SAP, which prunes networks based on the spectrum of NTK.
- We introduce a multi-sampling formulation which uses different weight configurations to better capture the expected behavior of pruned neural networks. A "*new-input-new-weight*" (NINW) trick is leveraged to reduce the computational cost, and may be of independent interest.
- Empirically, we show that NTK-SAP, as a data-agnostic foresight pruning method, achieves state-of-the-art performance in multiple settings.

## 2 RELATED WORK AND BACKGROUND

**Pruning after training (Post-hoc pruning).** Post-hoc pruning can be dated back to the 1980s (Janowsky, 1989; Mozer & Smolensky, 1989) and they usually require multiple rounds of train-prune-retrain procedure (Han et al., 2015; LeCun et al., 1989). Most of these pruning methods use

---

[1]Please see Appendix B.2 for more discussions from an empirical perspective.
[2]This is a subtle point, and we refer the readers to Appendix B.1 for more discussions.

magnitude (Han et al., 2015), Hessian (LeCun et al., 1989; Hassibi & Stork, 1992; Molchanov et al., 2019) and other information (Dai et al., 2018; Guo et al., 2016; Dong et al., 2017; Yu et al., 2018) to define the scoring metrics which determine which weights to prune. Some other recent works (Verma & Pesquet, 2021) identify sparse networks through iterative optimization. Most post-hoc pruning methods intend to reduce the inference time, with the exception of lottery ticket pruning (Frankle & Carbin, 2018; Renda et al., 2020) which prunes the networks for the purpose of finding a trainable sub-network.

**Pruning during training.** Another class of pruning algorithms gradually sparsifies a dense neural network during training. Some works use explicit $\ell_0$ (Louizos et al., 2017) or $\ell_1$ (Wen et al., 2016) regularization to learn a sparse solution. Other works introduce trainable masks (Liu et al., 2020; Savarese et al., 2020; Kang & Han, 2020; Kusupati et al., 2020; Srinivas et al., 2017; Xiao et al., 2019) or use projected gradient descent (Zhou et al., 2021) to specifically learn the connections to be remained. DST methods (Mocanu et al., 2018; Bellec et al., 2017; Mostafa & Wang, 2019; Dettmers & Zettlemoyer, 2019; Liu et al., 2021b; Evci et al., 2020; Liu et al., 2021c;a) train the neural network with a fixed parameter count budget while gradually adjust the parameters throughout training. Moreover, another line of work (You et al., 2019) tries to identify the winning ticket at early epochs. These methods can reduce some training time, but often do not reduce the memory cost of training.

**Pruning before training (foresight pruning).** SNIP (Lee et al., 2018) is arguably the first foresight pruning method. Later, GraSP (Wang et al., 2020) uses Hessian-gradient product to design the pruning metric. To address the layer collapse issue of SNIP and GraSP at high sparsity ratios, Synflow (Tanaka et al., 2020) prunes the network iteratively in a data-independent manner. FORCE and Iterative SNIP (de Jorge et al., 2020a) also intend to improve the performance of SNIP at high sparsity ratios. Lee et al. (2019b) uses orthogonal constraint to improve signal propagation of networks pruned by SNIP. Neural tangent transfer (Liu & Zenke, 2020) explicitly lets subnetworks learn the full empirical neural tangent kernel of dense networks. PropsPr (Alizadeh et al., 2022) also mentions the "shortsightedness" of existing foresight pruning methods and proposes to use meta-gradients to utilize the information of the first few iterations.

## 3 GENERAL PIPELINE FOR FORESIGHT PRUNING METHODS

In this section, we review the formulation of pruning-at-initialization problem and revisit several existing foresight pruning methods. Here we consider Synflow (Tanaka et al., 2020), GraSP (Wang et al., 2020), SNIP (Lee et al., 2018) and Iterative SNIP (de Jorge et al., 2020b). When given a dataset $\mathcal{D} = (\mathbf{x}_i, \mathbf{y}_i)_{i=1}^n$ and an untrained neural network $f(\cdot; \boldsymbol{\theta})$ parameterized by $\boldsymbol{\theta} = \{\theta^1, \theta^2, \cdots, \theta^p\} \in \mathbb{R}^p$, we are interested in finding a binary mask $\mathbf{m} = \{m^1, m^2, \cdots, m^p\} \in \{0, 1\}^p$ which minimizes the following equation:

$$\min_{\mathbf{m}} \mathcal{L}(\mathcal{A}(\boldsymbol{\theta}_0, \mathbf{m}) \odot \mathbf{m}; \mathcal{D}) = \min_{\mathbf{m}} \frac{1}{n} \sum_{i=1}^n \mathcal{L}(f(\mathbf{x}_i; \mathcal{A}(\boldsymbol{\theta}_0, \mathbf{m}) \odot \mathbf{m}), \mathbf{y}_i) \tag{1}$$

$$\text{s.t.} \quad \mathbf{m} \in \{0, 1\}^p, \quad \|\mathbf{m}\|_0/p \le d = 1 - k$$

where $k$ and $d = 1 - k$ are the target sparsity and density, $\boldsymbol{\theta}_0$ is an initialization point which follows a specific weight initialization method $\mathcal{P}$ (e.g. Xavier initialization), $\odot$ denotes the Hadamard product, and $\mathcal{A}$ denotes a training algorithm (e.g. Adam) which takes in mask $\mathbf{m}$ as well as initialization $\boldsymbol{\theta}_0$, and outputs trained weights at convergence $\boldsymbol{\theta}_{\text{final}} \odot \mathbf{m}$.

Since directly minimizing Equation (1) is intractable, existing foresight pruning methods assign each mask and corresponding weight at initialization $\theta_0^j$ with a saliency score $S(\theta_0^j)$ in the following generalized form:

$$S(m^j) = S(\theta_0^j) = \frac{\partial \mathcal{I}}{\partial m^j} = \frac{\partial \mathcal{I}}{\partial \theta_0^j} \cdot \theta_0^j \tag{2}$$

where $\mathcal{I}$ is a function of parameters $\boldsymbol{\theta}_0$ and corresponding mask $\mathbf{m}$. Intuitively, Equation (2) measures the influence of removing connection $m^j$ on $\mathcal{I}$. Once the saliency score for each mask is computed, we will only keep the top-$d$ masks. Specifically, the masks with the highest saliency scores will be retained ($m^j = 1$), while other masks will be pruned ($m^j = 0$).

**SNIP.** SNIP (Lee et al., 2018) defines $S_{\text{SNIP}}(m^j) = \left| \frac{\partial \mathcal{L}(\boldsymbol{\theta}_0 \odot \boldsymbol{m}; D)}{\partial \theta_0^j} \cdot \theta_0^j \right|$, where $D \subset \mathcal{D}$ is a subset of the whole training set. SNIP prunes the whole network in one round.

**Iterative SNIP.** The saliency score of Iterative SNIP[3] (de Jorge et al., 2020b) remains the same as the original SNIP. However, multiple rounds of pruning are performed.

**GraSP.** GraSP (Wang et al., 2020) defines $S_{\text{GraSP}}(m^j) = - \left[ H(\boldsymbol{\theta}_0 \odot \boldsymbol{m}; D) \frac{\partial \mathcal{L}(\boldsymbol{\theta}_0 \odot \boldsymbol{m}; D)}{\partial \boldsymbol{\theta}_0} \right]^j \cdot \theta_0^j$, where $H$ is the Hessian matrix. Additionally, GraSP is also a single round foresight pruning method.

**Synflow.** Synflow (Tanaka et al., 2020) uses $S_{\text{Synflow}}(m^j) = \left| \frac{\partial \sum_k f(\mathbb{1}; |\boldsymbol{\theta}_0 \odot \boldsymbol{m})_k}{\partial |\theta_0^j|} \cdot |\theta_0^j| \right|$ as saliency score, where $f_k$ denotes the $k$-th output logit, $\mathbb{1}$ denotes an input full of 1s, and $f(\cdot; |\theta_0^j|)$ means replacing all the parameters of the neural network by their absolute values. Similar to Iterative SNIP, Synflow is also an iterative pruning method but in a data-agnostic manner.

## 4 REVIEW OF NEURAL TANGENT KERNEL

Recent works (Jacot et al., 2018; Lee et al., 2019a; Arora et al., 2019; Xiao et al., 2020) study the training dynamics of large DNNs under gradient descent. Formally, let $\mathcal{D} = \{(\mathbf{x}_i, \mathbf{y}_i)\}_{i=1}^n \subset \mathbb{R}^{n_0} \times \mathbb{R}^k$ denote the training set and let $\mathcal{X} = \{\mathbf{x} : (\mathbf{x}, \mathbf{y}) \in \mathcal{D}\}$, $\mathcal{Y} = \{\mathbf{y} : (\mathbf{x}, \mathbf{y}) \in \mathcal{D}\}$ denote inputs and labels, respectively. Consider a neural network $f(\cdot; \boldsymbol{\theta})$ parameterized by weights $\boldsymbol{\theta}$. Let $\mathcal{L}(\boldsymbol{\theta}) = \sum_{(\mathbf{x}, \mathbf{y}) \in (\mathcal{X}, \mathcal{Y})} \ell(f(\mathbf{x}; \boldsymbol{\theta}), \mathbf{y})$ denote the empirical loss where $\ell$ is the loss function. Thus, $f(\mathcal{X}; \boldsymbol{\theta}) \in \mathbb{R}^{kn \times 1}$ is the concatenated vector of predictions for all training inputs. Under continuous time gradient descent with learning rate $\eta$, the evolution of the parameters $\boldsymbol{\theta}_t$, where we emphasize the dependence of $\boldsymbol{\theta}$ on time $t$, and the training loss of the neural network $\mathcal{L}$ can be expressed (Lee et al., 2019a) as

$$\dot{\boldsymbol{\theta}}_t = -\eta \nabla_{\boldsymbol{\theta}_t} f(\mathcal{X}; \boldsymbol{\theta}_t)^T \nabla_{f(\mathcal{X}; \boldsymbol{\theta}_t)} \mathcal{L} \tag{3}$$

$$\dot{\mathcal{L}} = \nabla_{f(\mathcal{X}; \boldsymbol{\theta}_t)} \mathcal{L}^T \nabla_{\boldsymbol{\theta}_t} f(\mathcal{X}; \boldsymbol{\theta}_t) \dot{\boldsymbol{\theta}}_t = -\eta \nabla_{f(\mathcal{X}; \boldsymbol{\theta}_t)} \mathcal{L}^T \hat{\boldsymbol{\Theta}}_t(\mathcal{X}, \mathcal{X}) \nabla_{f(\mathcal{X}; \boldsymbol{\theta}_t)} \mathcal{L} \tag{4}$$

where we obtain the NTK at time $t$:

$$\hat{\boldsymbol{\Theta}}_t(\mathcal{X}, \mathcal{X}; \boldsymbol{\theta}_t) \triangleq \nabla_{\boldsymbol{\theta}_t} f(\mathcal{X}; \boldsymbol{\theta}_t) \nabla_{\boldsymbol{\theta}_t} f(\mathcal{X}; \boldsymbol{\theta}_t)^T \in \mathbb{R}^{kn \times kn}. \tag{5}$$

It is proven that when the width of the neural network becomes sufficiently large (Jacot et al., 2018; Arora et al., 2019), the fixed-weight-NTK converges to a deterministic matrix $\boldsymbol{\Theta}(\mathcal{X}, \mathcal{X})$, which we term analytic NTK. The analytic NTK remains asymptotically constant throughout training, i.e. $\boldsymbol{\Theta}_t(\mathcal{X}, \mathcal{X}) \approx \boldsymbol{\Theta}_0(\mathcal{X}, \mathcal{X})$, $\forall t \geq 0$.

**The condition number of the NTK is related to neural network optimization.** The condition number of the NTK is known to be important in neural network optimization. Specifically, let the eigenvalue decomposition of the NTK at initialization be $\boldsymbol{\Theta}_0(\mathcal{X}, \mathcal{X}) = U^T \Lambda U$ where $\Lambda = \text{diag}(\lambda_1, \cdots, \lambda_{kn})$ is the diagonal matrix containing all the eigenvalues of the NTK.

For large-width networks, the mean prediction of the neural network in the eigenbasis of NTK can be approximately described as $(U \mathbb{E}[f(\mathcal{X})])_i = (\mathbf{I} - e^{-\eta \lambda_i t})(U \mathcal{Y})_i$ (Xiao et al., 2020). Therefore, when using the largest possible learning rate $\eta \sim 2/\lambda_1$ (Lee et al., 2019a), the convergence rate of the smallest eigenvalue is related to $\lambda_1/\lambda_{kn}$, which is the condition number of the NTK. Empirically, the condition number of the NTK has been successfully used to identify promising architectures in the neural architecture search (NAS) field (Chen et al., 2021; Wang et al., 2022).

**The entire spectrum of the NTK is a better measure.** However, the smallest and the largest eigenvalues only provide very limited information on the optimization property of the DNNs. In the classical optimization theory, the convergence speed depends on all eigenvalues of the Hessians, as different eigenvalues control the convergence of different eigen-modes. Similarly, for neural networks, a few works (Kopitkov & Indelman, 2020; Su & Yang, 2019) have shown that the convergence speed depends on eigenvalues other than the smallest eigenvalue. In this sense, compared to only considering the condition number of the NTK, controlling the entire eigenspectrum may be a better way to ensure the satisfactory optimization performance of DNNs.

---

[3]We do not consider FORCE in this work since we find empirically that FORCE is less stable than Iterative SNIP. See Appendix F for a detailed comparison.

## 5    OUR METHOD

### 5.1    ACCOUNT FOR THE EFFECTS OF THE ENTIRE EIGENSPECTRUM WITH TRACE NORM

As discussed in Section 4, the eigenspectrum of the NTK captures the training dynamics of large DNNs, and thus we intend to use the NTK spectrum to guide foresight pruning. More specifically, we would like to keep the spectrum of the pruned network close to that of the large network, so that the pruned network may perform similarly to the large network.

As discussed in Section 3, the saliency-score-based method requires using a single metric to capture the optimization performance. This single metric is often a scalar, and thus we want to generate a scalar to capture the information of the whole spectrum. A very natural choice is to use the average value of all eigenvalues. In fact, Pennington et al. (2017; 2018); Shu et al. (2021) indeed analyze the optimization property of DNNs through the average eigenvalues of the NTK (or a related matrix called input-output Jacobian) explicitly or implicitly. Additionally, using the average eigenvalues have also a few good properties which make it more suitable than other metrics for pruning:

**Computational efficiency.** To begin with, the computational cost of computing all the eigenvalues would be too expensive. On the contrary, trace norm can be efficiently computed, which we will show in Section 5.2.

**Potentially preserve the entire spectrum.** As indicated in Section 3, the saliency score for each weight measures the change of the metric (here it is the trace norm of the NTK) when removing the weight. More precisely, removing weights with a smaller saliency score is less likely to induce significant change to the eigenvalue distribution. Hence, using the NTK trace norm as the indicator is beneficial in maintaining the entire spectrum.

### 5.2    APPROXIMATING THE TRACE NORM OF THE NTK

**Nuclear norm as a proxy of the whole spectrum.** The NTK is a symmetric matrix. Hence, the trace of the NTK is equivalent to the squared Frobenius norm of parameter-output Jacobian, i.e.

$$\|\boldsymbol{\Theta}_0\|_* = \|\boldsymbol{\Theta}_0\|_{\mathrm{tr}} = \|\nabla_{\boldsymbol{\theta}_0} f(\mathcal{X}; \boldsymbol{\theta}_0)\|_F^2. \tag{6}$$

**Finite difference approximation.** Computing the Frobenius norm of the Jacobian matrix $\nabla_{\boldsymbol{\theta}_0} f(\mathcal{X}; \boldsymbol{\theta}_0) \in \mathbb{R}^{kn \times p}$ is memory and computation heavy. More specifically, a naive solution that constructs the Jacobian matrix explicitly for a batch of training data may require at least $k$ rounds of backward propagation, where $k$ is as large as 1000 for ImageNet classification task. So approximation is needed.

One possible way is to lower-bound it using loss gradient with respect to weights $\|\nabla_{\boldsymbol{\theta}_0}\mathcal{L}\|_2^2$. Such a method has been studied in FORCE (de Jorge et al., 2020b) but found to be inferior to Iterative SNIP.

Alternatively, we use the finite difference expression $\frac{1}{\epsilon}\mathbb{E}_{\Delta\boldsymbol{\theta}\sim\mathcal{N}(\mathbf{0},\epsilon\mathbf{I})}\left[\|f(\mathcal{X}; \boldsymbol{\theta}_0) - f(\mathcal{X}; \boldsymbol{\theta}_0 + \Delta\boldsymbol{\theta})\|_2^2\right]$ to approximate the Jacobian norm. Please see Appendix O for the justification.

**Approximating the training set with pruning set.** We use the following finite approximation expression to define a saliency score:

$$S_{\mathrm{NTK\text{-}SAP}}(m^j) = \left|\frac{\partial\mathbb{E}_{\Delta\boldsymbol{\theta}\sim\mathcal{N}(\mathbf{0},\epsilon\mathbf{I})}\left[\|f(\mathbf{X}_D; \boldsymbol{\theta}_0 \odot \mathbf{m}) - f(\mathbf{X}_D; (\boldsymbol{\theta}_0 + \Delta\boldsymbol{\theta}) \odot \mathbf{m})\|_2^2\right]}{\partial m^j}\right| \tag{7}$$

where we use the inputs of the pruning dataset $\mathbf{X}_D$ to approximate the whole training inputs $\mathcal{X}$. Our foresight pruning method will use the same framework shown in Section 3, which prunes the weights with the least saliency scores defined in Equation (7). Nevertheless, a few more tricks are needed to make this method practical, as discussed next.

### 5.3    NTK-SAP: MULTI-SAMPLING FORMULATION

The last subsection computes the saliency score using the fixed-weight-NTK. We think the spectrum of analytic NTK may serve as a performance indicator of a certain architecture. While a single fixed-weight-NTK can be viewed as an approximation of the analytic NTK, the expectation

of fixed-weight-NTKs (expecation over random draw of weight configurations) shall be a better approximation. Further, to approximate the expectation, we sample a few independent realization of weight configurations and compute the average of their fixed-weight-NTKs. This helps explore the parameter space and better capture the expected behavior of the pruned neural networks.

Specifically, with $R$ random weight configurations $\boldsymbol{\theta}_{0,r} \overset{\text{iid}}{\sim} \mathcal{P}$, we use the following stabilized version of the saliency score:

$$S_{\text{NTK-SAP}}(m^j) = \left| \frac{\partial \frac{1}{R} \sum_{r=1}^{R} \mathbb{E}_{\Delta\boldsymbol{\theta} \sim \mathcal{N}(\mathbf{0},\epsilon\mathbf{I})} \left[ \|f(\mathbf{X}_D; \boldsymbol{\theta}_{0,r} \odot \mathbf{m}) - f(\mathbf{X}_D; (\boldsymbol{\theta}_{0,r} + \Delta\boldsymbol{\theta}) \odot \mathbf{m})\|_2^2 \right]}{\partial m^j} \right|. \tag{8}$$

There is a major difference between the proposed saliency score and most existing foresight pruning scores: our score is weight-agnostic, i.e., it is not defined by a specific random draw of weight configuration. In other words, the score is mainly determined by the mask structure rather than specific weights.[4]

**New-input-new-weight (NINW) trick.** It seems that using multiple weight configurations will cause a large increase in the computational cost. Nevertheless, by utilizing the fact that the training subset for pruning $D = (\mathbf{x}_i, \mathbf{y}_i)_{i=1}^{|D|} \subset \mathcal{D}$ is usually fed into neural networks in multiple batches, we further approximate Equation (8) with NINW trick as

$$S_{\text{NTK-SAP}}(m^j) = \left| \frac{\partial \frac{1}{|D|} \sum_{i=1}^{|D|} \mathbb{E}_{\Delta\boldsymbol{\theta} \sim \mathcal{N}(\mathbf{0},\epsilon\mathbf{I})} \left[ \|f(\mathbf{x}_i; \boldsymbol{\theta}_{0,i} \odot \mathbf{m}) - f(\mathbf{x}_i; (\boldsymbol{\theta}_{0,i} + \Delta\boldsymbol{\theta}) \odot \mathbf{m})\|_2^2 \right]}{\partial m^j} \right|. \tag{9}$$

**Replacing pruning set with random input.** Furthermore, we find that replacing the training subset $D$ with standard Gaussian noise $\mathcal{Z} \sim \mathcal{N}(\mathbf{0}, \mathbf{I})$ can generate comparable performance, which makes NTK-SAP purely data-agnostic. As for Gassuan perturbation, ideally, using multiple draws will indeed give a more accurate approximation. However, we empirically find that a single draw of Gaussian perturbation is good enough to generate satisfactory performance. Hence, by using one random draw of $\Delta\boldsymbol{\theta}$ we have the final data-agnostic NTK-SAP with its saliency score

$$S_{\text{NTK-SAP}}(m^j) = \left| \frac{\partial \frac{1}{|D|} \sum_{i=1}^{|D|} \left[ \|f(\mathbf{z}_i; \boldsymbol{\theta}_{0,i} \odot \mathbf{m}) - f(\mathbf{z}_i; (\boldsymbol{\theta}_{0,i} + \Delta\boldsymbol{\theta}_i) \odot \mathbf{m})\|_2^2 \right]}{\partial m^j} \right| \tag{10}$$

where $\boldsymbol{\theta}_{0,i} \overset{\text{iid}}{\sim} \mathcal{P}$, $\Delta\boldsymbol{\theta}_i \overset{\text{iid}}{\sim} \mathcal{N}(\mathbf{0}, \epsilon\mathbf{I})$ and $\mathbf{z}_i \overset{\text{iid}}{\sim} \mathcal{N}(\mathbf{0}, \mathbf{I})$.

In addition, to potentially avoid layer collapse, we also iteratively prune neural networks like Synflow in $T$ rounds.[5] The major steps of the NTK-SAP algorithm are given in Algorithm 1.

**Computational cost.** We compare the computational cost of NTK-SAP with Iterative SNIP as they are both iterative methods and use multi-batch inputs. Since the overhead of the reinitialization process is negligible, NTK-SAP needs roughly double the computational cost of Iterative SNIP due to two forward passes with the computational graph in the formula. However, one should note that NTK-SAP is a data-agnostic method that can be computed beforehand, and the computation time can be reduced with a smaller $T$; thus, the computational overhead is reasonable. Effects of $T$, a detailed comparison of pruning time, and FLOPs comparison can be found in Section 6.4, Appendix D.1, and Appendix D.2, respectively.

# 6 EXPERIMENTS

In this section, we empirically evaluate the effectiveness of our proposed method, NTK-SAP (red). We compare with two baseline methods random pruning (gray) and magnitude pruning (magenta), as

---

[4]Please refer to Appendix M for a discussion on whether a good pruning solution should be weight-agnostic.
[5]Please refer to Appendix N for a discussion on why NTK-SAP could potentially avoid layer collapse.

---

**Algorithm 1** Neural Tagent Kernel Spectrum-Aware Pruning (NTK-SAP)

---

**Require:** Dense network $f(\cdot; \boldsymbol{\theta} \odot \mathbf{m})$, final density $d$, iteration steps $T$, batch size of noise input $B$, perturbation hyperparameter $\epsilon$

1: Initialize $\mathbf{m} = \mathbb{1}$
2: **for** $t$ in $[1, \cdots, T]$ **do**
3:      **for** $i$ in $[1, \cdots, B]$ **do**
4:          Reinitialize neural network with $\boldsymbol{\theta}_{0,i} \overset{\text{iid}}{\sim} \mathcal{P}$
5:          Sample noise input $\mathbf{z}_i \overset{\text{iid}}{\sim} \mathcal{N}(\mathbf{0}, \mathbf{I})$ and parameter perturbation $\Delta\boldsymbol{\theta}_i \overset{\text{iid}}{\sim} \mathcal{N}(\mathbf{0}, \epsilon\mathbf{I})$
6:          Evaluate saliency scores $[\mathcal{S}(\mathbf{m})]_i$
7:      **end for**
8:      Calculate $\mathcal{S}(\mathbf{m}) = \left| \sum_{i=1}^{B} [\mathcal{S}(\mathbf{m})]_i \right|$
9:      Compute $(1 - d^{\frac{t}{T}})$ percentile of $\mathcal{S}(\mathbf{m})$ as threshold $\tau$
10:     Update mask $\mathbf{m} \leftarrow \mathbf{m} \odot (\mathcal{S}(\mathbf{m}) > \tau)$
11: **end for**
12: **return** Final mask $\mathbf{m}$

---

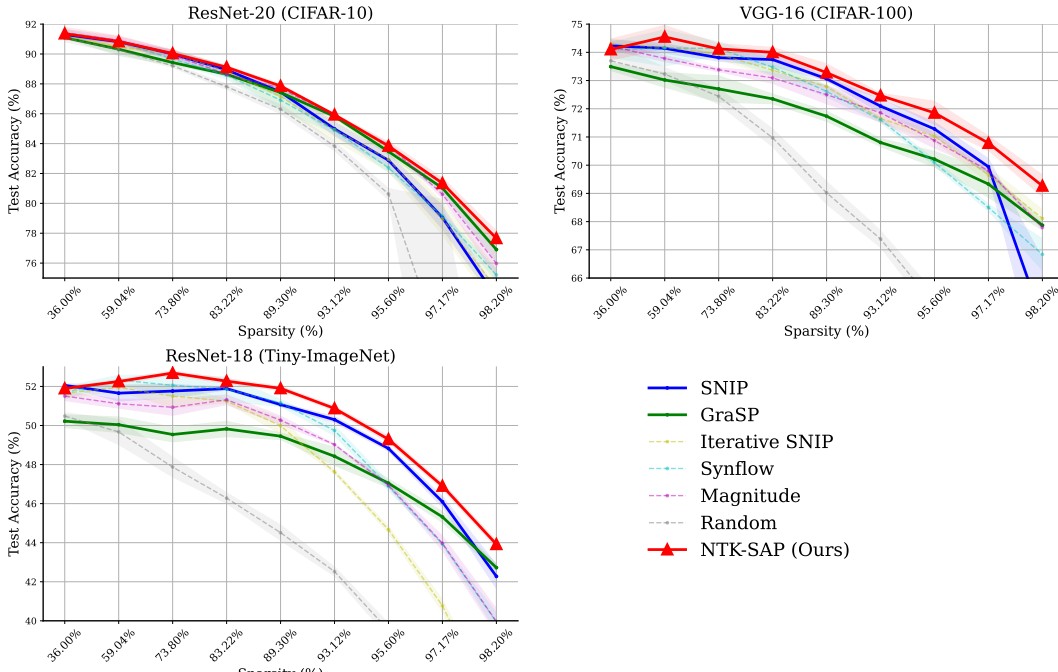

Figure 1: Performance of NTK-SAP against other foresight pruning methods. Results are averaged over 3 random seeds, and the shaded areas denote the standard deviation. We highlight SNIP and GraSP as they perform best among baselines on mild and extreme density ratios, respectively.

well as foresight pruning algorithms SNIP (blue) (Lee et al., 2018), Iterative SNIP (yellow) (de Jorge et al., 2020b), GraSP (green) (Wang et al., 2020) and Synflow (cyan) (Tanaka et al., 2020) across multiple architectures and datasets.

**Target datasets, models, and sparsity ratios.** We develop our code based on the original code of Synflow (Tanaka et al., 2020). On smaller datasets, we consider ResNet-20 (He et al., 2016) on CIFAR-10, VGG-16 (Simonyan & Zisserman, 2014) on CIFAR-100 (Krizhevsky et al., 2009) and ResNet-18 on Tiny-ImageNet. For ImageNet (Deng et al., 2009) experiments, we consider ResNet-18 and ResNet-50. For the training protocols, we roughly follow the settings in Frankle et al. (2020b). Details of training hyperparameters can be found in Appendix A. We divide the sparsity ratios into three ranges: trivial sparsity ratios (36.00%, 59.04% and 73.80%), mild sparsity ratios (83.22%, 89.30% and 93.12%) and extreme sparsity ratios (95.60%, 97.17% and 98.20%).

**Implementation details.** Following the original implementation, SNIP, GraSP, and Iterative SNIP are run by setting BatchNorm layers as train mode while Synflow metric is evaluated under evaluation mode. The number of training samples for pruning we use is ten times of the number of classes as suggested by the original implementations of SNIP and GraSP. For iterative methods, including Iterative SNIP and Synflow, we prune all the weights in 100 rounds in an exponential schedule following Tanaka et al. (2020); Frankle et al. (2020b). Further, following Frankle et al. (2020b), Kaiming normal initialization is used for all models. We prune networks using a batch size of 256 for CIFAR-10/100 and Tiny-ImageNet datasets and a batch size of 128 for ImageNet experiments.

For NTK-SAP, we use the same number of Gaussian noise batches as in Iterative SNIP and prunes neural network progressively in $T = 100$ rounds for CIFAR-100 and Tiny-ImageNet. Since the pruning dataset for CIFAR-10 only contains 1 batch of the training sample, we alternatively sample $R = 5$ independent initialization points and prune progressively in $T = 20$ rounds, keeping the overall computational cost roughly the same. More details can be found in Appendix A.

## 6.1 RESULTS ON CIFAR AND TINY-IMAGENET

In Figure 1 we evaluate NTK-SAP using ResNet-20 on CIFAR-10, VGG-16 on CIFAR-100 and ResNet-18 on Tiny-ImageNet. It can be observed that NTK-SAP is competitive for all sparsity ratios.

**Trivial and mild sparsity ratios.** For trivial sparsity ratios, all methods show good performance except GraSP, magnitude, and random pruning. On ResNet-20, all methods are similar, while random pruning and GraSP show slightly worse performance. On VGG-16 (ResNet-18), similar trend can be observed while the performance gap is larger. NTK-SAP shows consistent good performance, especially on ResNet-18 (TinyImageNet).

For mild sparsity ratios, NTK-SAP is the only method that performs well across all datasets and models. On ResNet-20, SNIP and Iterative SNIP take the lead with NTK-SAP while suffering from a performance drop starting at 93.12% sparsity. Magnitude pruning and GraSP have similar performance to NTK-SAP after 89.30%. On VGG-16 and ResNet-18, NTK-SAP and SNIP remain dominant, while NTK-SAP outperforms SNIP.

**Extreme sparsity ratios.** NTK-SAP's superior performance can be seen at extreme sparsity ratios, especially 98.20%. On the CIFAR-10 dataset, GraSP and magnitude pruning are competitive as in matching sparsity ratios. On VGG-16, SNIP first dominates with NTK-SAP and then becomes much inferior. At the sparsest level 98.20%, magnitude, Iterative SNIP, and GraSP show similar results, about 1 percentage point lower than NTK-SAP.

## 6.2 RESULTS ON LARGER DATASET

We further evaluate the performance of NTK-SAP on more difficult tasks: we run experiments on the ImageNet dataset using ResNet-18 and ResNet-50 with pruning ratios {89.26%, 95.60%}.[6] We compare NTK-SAP against Synflow, SNIP, Iterative SNIP, GraSP, Magnitude, and Random pruning.

The results are shown in Table 1. We find that SNIP, which performs reasonably well for smaller datasets, is no longer competitive. For ResNet-50, NTK-SAP beats other pruning methods with margins at least 0.5% and 1% for sparsity ratio 89.26% and 95.60%, respectively. For ResNet-18, NTK-SAP slightly outperforms magnitude pruning under 89.26% sparsity, and GraSP follows. For sparsity ratio 95.60 %, GraSP is better than magnitude pruning while NTK-SAP still takes the lead.

One should note the surprisingly good performance of magnitude pruning. We hypothesize that magnitude pruning chooses an appropriate layer-wise sparsity ratio since Kaiming normal initialization already gives weights different variances depending on the number of parameters of different layers. Thus, layers with more parameters will be more likely pruned.

## 6.3 AN OVERALL COMPARISON WITH SNIP AND GRASP

In this section, we explicitly compare NTK-SAP with two competitive baseline methods in detail, i.e., SNIP and GraSP, since there is no state-of-the-art foresight pruning algorithms (Frankle et al., 2020b).

---

[6]We choose these sparsity ratios as they include matching and extreme sparsity ratios and close to {90%, 95%} as used by Iterative SNIP (de Jorge et al., 2020b) and ProsPr (Alizadeh et al., 2022).

Table 1: Test performance of foresight pruning methods on the ImageNet dataset. Best results are in **bold**; second-best results are underlined.

| Network | ResNet-18 | | ResNet-50 | |
|---|---|---|---|---|
| Sparsity percentage | 89.26% | 95.60% | 89.26% | 95.60% |
| (Dense Baseline) | 69.98 | | 76.20 | |
| Synflow | 57.30 | 45.65 | 66.81 | 58.88 |
| SNIP | 57.41 | 45.04 | 60.98 | 40.69 |
| Iterative SNIP | 52.97 | 37.44 | 52.53 | 36.82 |
| GraSP | 58.16 | 49.17 | 67.74 | 59.73 |
| Magnitude | 58.75 | 48.50 | 66.80 | 43.79 |
| Random | 54.48 | 42.80 | 65.30 | 56.53 |
| **Our method** | **58.87** | **49.43** | **68.28** | **60.79** |

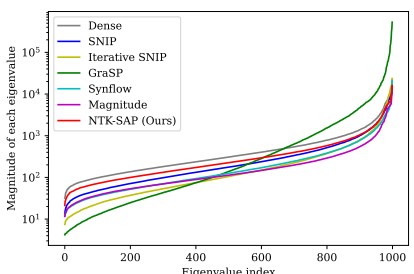

Figure 2: Eigenvalue distribution of the fixed-weight-NTK on ResNet-20 (CIFAR-10) at sparsity ratio $73.80\%$.

We highlight both baseline algorithms in Figure 1. Specifically, SNIP is considered competitive as it generally performs well, especially for trivial and mild sparsity ratios. However, SNIP shows poor performance for ImageNet. On the contrary, GraSP indeed shows promising performance on the ImageNet dataset and extreme sparsity ratios on smaller datasets, while it suffers from poor performance on trivial and mild sparsity ratios of smaller datasets. In this sense, NTK-SAP is the first foresight pruning algorithm that works consistently well on all datasets.

### 6.4 MORE ABLATION STUDIES

We finally evaluate the performance of NTK-SAP when varying the number of iterative steps ($T$), which can give a better understanding of how iterative pruning helps NTK-SAP achieve better performance at higher sparsity ratios. We repeat experiments of Section 6.1 with varying $T$. Results show that increasing $T$ consistently boosts the performance of NTK-SAP. Moreover, such an increase saturates when $T$ is roughly $1/5$ of the adopted value in Section 6.1. Hence, the computational cost of NTK-SAP can be further decreased if we use a smaller value of $T$. Please refer to Appendix C for more details. We also conduct an ablation study on the effects of perturbation hyperparameter $\epsilon$. Results show that NTK-SAP is robust to the choices of $\epsilon$. See Appendix I for more details.

### 6.5 THE EIGENSPECTRUM OF THE FIXED-WEIGHT-NTK AFTER PRUNING

To visualize the effectiveness of our proposed NTK-SAP method, we show eigenspectrum of pruned subnetwork on CIFAR-10 dataset with ResNet-20 at sparsity ratios $73.80\%$. Figure 2 shows that NTK-SAP successfully maintain the eigenspectrum of the fixed-weight-NTK compared to other methods. See Appendix G.1 for the eigenspectrum of the fixed-weight-NTK for other sparsity ratios.

## 7 CONCLUSION

Pruning at initialization is appealing compared to pruning-after-training algorithms since it circumvents the need for training to identify the masks. However, most of the existing foresight pruning methods do not capture the training dynamics after pruning. We argue that such an issue can be mitigated by using NTK theory. Hence, we propose NTK-SAP, which iteratively prunes neural networks without any data. Additionally, NTK-SAP pays attention to multiple random draws of initialization points. Empirically, our method is competitive in all datasets and models studied in this work. We consider that our algorithm can encourage more works to apply theoretical findings to empirical applications. Additionally, our novel way of combining multiple initialization configurations may further encourage future work to investigate the similarity of pruning and NAS just like Liu et al. (2018); Abdelfattah et al. (2021).

The limitations of the work include the relatively heavy computation nature of iterative pruning methods compared to single-shot pruning methods. We also want to emphasize the great significance of pruning methods in reducing the energy cost of training and applying deep learning models.

## 8 ACKNOWLEDGEMENT

This work utilizes resources supported by the National Science Foundation's Major Research Instrumentation program, grant No.1725729 (Kindratenko et al., 2020), as well as the University of Illinois at Urbana-Champaign. This paper is supported in part by Hetao Shenzhen-Hong Kong Science and Technology Innovation Cooperation Zone Project (No.HZQSWS-KCCYB-2022046); University Development Fund UDF01001491 from the Chinese University of Hong Kong, Shenzhen; Guangdong Key Lab on the Mathematical Foundation of Artificial Intelligence, Department of Science and Techonology of Guangdong Province.

We would like to thank the program chairs and the area chairs for handling our paper. We would also like to thank the reviewers for their time and efforts in providing constructive suggestions on our paper.

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

OVERVIEW OF THE APPENDIX

The Appendix is organized as follows:

- Appendix A introduces the general experimental setup.
- Appendix B provides intuitions on our multi-sampling formulation.
- Appendix C shows an ablation study on the number of iterative steps.
- Appendix D provides a detailed comparison of computational costs of different foresight pruning methods.
- Appendix E provides a further comparison between foresight pruning methods and the early pruning baseline.
- Appendix F provides a comparison of SNIP variants.
- Appendix G shows eigenspectrum of pruned neural networks.
- Appendix H provides full plots of the main experiments.
- Appendix I provides ablation study on the perturbation hyper-parameter $\epsilon$.
- Appendix J provides an overview of foresight pruning methods considered in the paper.
- Appendix K provides a comparison between neural tangent transfer and NTK-SAP.
- Appendix L provides a comparison between ProsPr and NTK-SAP.
- Appendix M includes a discussion on the weight-agnostic property of NTK-SAP.
- Appendix N includes a discussion on why NTK-SAP can potentially avoid layer collapse.
- Appendix O justifies the finite difference formulation used in Section 5.2.

## A    EXPERIMENTAL SETUP

Our code is based on the original code of Synflow.[7]  The original code does not contain license information. But we obtained approval via email. The authors grant free use for academic purposes.

### A.1    ARCHITECTURE DETAILS

The networks architectures are as follows:

- For the ResNet-20 model on CIFAR-20 and VGG-16 (with BatchNorm) model on CIFAR-100, we use the default ones used in Synflow (Tanaka et al., 2020) experiments. Lottery ticket experiments (Frankle & Carbin, 2018; Renda et al., 2020; Frankle et al., 2020a) use the same architectures.
- For ResNet-18 and ResNet-50, we use the ImageNet version of ResNet models.[8] We use the `torchvision` implementations.

### A.2    DATASETS

We use CIFAR-10, CIFAR-100, Tiny-ImageNet and ImageNet. They do not contain personally identifiable information or offensive content.

- CIFAR-10 and CIFAR-100 datasets are augmented by per-channel normalization, applying random cropping ($32 \times 32$, padding 4), and horizontal flip.
- Tiny-ImageNet is augmented by per-channel normalization, choosing a patch with a random aspect ratio between 0.8 and 1.25 and a random scale between 0.1 and 1.0, cropping to $64 \times 64$, and random horizontal flip.
- ImageNet dataset is augmented by per-channel normalization, applying random resized cropping to $224 \times 224$, and random horizontal flip.

---

[7]https://github.com/ganguli-lab/Synaptic-Flow.
[8]https://github.com/weiaicunzai/pytorch-cifar100

## A.3 TRAINING DETAILS

See Table 2 for training hyper-parameters. These hyper-parameters are modified from the ones used in Synflow (Tanaka et al., 2020) and Frankle et al. (2020b). In the following table, we use `Nesterov` optimizer to denote SGD optimizer with Nesterov momentum.

Table 2: Training hyper-parameters used in this work.

| Network | Dataset | Epochs | Batch | Optimizer | Momentum | LR | LR drop | Weight decay | Initilization |
|---------|---------|--------|-------|-----------|----------|-----|---------|--------------|---------------|
| ResNet-20 | CIFAR-10 | 160 | 128 | SGD | 0.9 | 0.1 | 10x at epochs 80, 120 | 1e-4 | Kaiming Normal |
| VGG-16 | CIFAR-100 | 160 | 128 | Nesterov | 0.9 | 0.1 | 10x at epochs 60, 120 | 5e-4 | Kaiming Normal |
| ResNet-18 | Tiny-ImageNet | 200 | 256 | SGD | 0.9 | 0.2 | 10x at epochs 100, 150 | 1e-4 | Kaiming Normal |
| ResNet-18 | ImageNet | 90 | 512 | SGD | 0.9 | 0.1 | 10x at epochs 30, 60, 80 | 1e-4 | Kaiming Normal |
| ResNet-50 | ImageNet | 90 | 448 | SGD | 0.9 | 0.1 | 10x at epochs 30, 60, 80 | 1e-4 | Kaiming Normal |

## A.4 PRUNING DETAILS

**Sparsity ratios**   The x-axis for Figure 1 is in the logarithmic scale of density (1-sparsity). More specifically, we obtain each sparsity ratio by removing 20% of the remaining parameters twice. For example, the first sparsity ratio is 1-0.8*0.8=0.36. We choose these sparsity ratios to evaluate mainly following the conventional practice used in Frankle et al. (2020b). The intuition is that for the first few rounds, the network is over-parameterized and a large portion of the weights can be removed.

**Choosing pruning samples.**   Different from GraSP implementation, which uses exactly 10 examples of each class, we follow the implementation of SNIP, Iterative SNIP, and Synflow to load multiple batches so that each class roughly has ten samples.

**Pruning batch size.**   To compute the saliency scores, we use a pruning batch size of 128 for ImageNet experiments and use 256 otherwise.

**BatchNorm layers.**   As pointed out by the original Synflow paper, BatchNorm layers are crucial for applying different pruning methods correctly. SNIP, Iterative SNIP, and GraSP prune networks under train mode, while Synflow uses evaluation mode. As for NTK-SAP, we compute the saliency score in a modified evaluation mode. More specifically, we first compute batch statistics without constructing a computational graph, and then use the same batch statistics to get an approximation of the NTK nuclear norm under evaluation mode.

**Pruning schedule for iterative methods.**   Following Synflow, all iterative methods, including Synflow, Iterative SNIP, and NTK-SAP, use an exponential pruning schedule. See Algorithm 1 for how to compute the pruning schedule.

**More details about NTK-SAP.**   For each round of pruning, we use 40 batches (compared to 79 batches for Iterative SNIP) of noise inputs with a batch size of 128 for ResNet-50 on the ImageNet dataset. For other settings, the number of Gaussian inputs for pruning we use is ten times the number of classes.

As for the number of pruning rounds $T$, we use 25 for all ImageNet experiments. And we use $T = 20$ rounds for ResNet-20 on CIFAR-10 with 5 batches each round (as the batch size is larger than the number of classes 10). For CIFAR-100 and Tiny-ImageNet datasets, we use $T = 100$ rounds.

For $\epsilon$, we use a value of $\epsilon = 10^{-6}$ for ResNet-18 on ImageNet and use a value of $\epsilon = 10^{-4}$ otherwise.

## A.5 HARDWARE

All of our experiments were run on NVIDIA V100s. Experiments on CIFAR-10/100 and Tiny-ImageNet datasets were run on a single GPU at a time. We use 2 and 4 GPUs for ResNet-18 and ResNet-50 in ImageNet experiments, respectively.

## B    Intuition of multi-sampling formulation

### B.1    Understanding multi-sampling formulation through the analytic NTK

In this section, we elaborate on our intuition of performing multi-sampling in NTK-SAP.

We still start our discussion with Equation 7. We note that Equation 7 alone is not enough to guarantee good performance, as there is actually a gap in applying NTK theory directly to pruning: in a real-world application, the fixed-weight-NTK may differ drastically from the analytic NTK.

**Standard expression of the analytic NTK.** We note that directly using the fixed-weight-NTK to approximate the analytic NTK can be written in the following mathematical formulation:

$$\Theta(\mathcal{X}, \mathcal{X}) = \lim_{n_l \to \infty, \forall l} \hat{\Theta}^{\{n_l\}}(\mathcal{X}, \mathcal{X}; \boldsymbol{\theta}) \tag{11}$$

where $\Theta$ is the analytic NTK, and $\hat{\Theta}^{\{n_l\}}(\cdot, \cdot; \boldsymbol{\theta})$ is the fixed-weight-NTK for width-$n_l$ neural networks with weight $\boldsymbol{\theta}$. A natural way to approximate the analytic NTK is to remove the "lim" operation and just use $\hat{\Theta}^{\{n_l\}}(\mathcal{X}, \mathcal{X}; \boldsymbol{\theta})$ for a given set of large enough width $\{n_l\}$.

**Two differences of the analytic NTK and the fixed-weight NTK.** We point out that besides the choice of widths, the choice of weights $\boldsymbol{\theta}$ also affects $\hat{\Theta}^{\{n_l\}}(\mathcal{X}, \mathcal{X}; \boldsymbol{\theta})$. In other words, $\Theta(\mathcal{X}, \mathcal{X})$ is both width-agnostic and weight-agnostic, while $\hat{\Theta}^{\{n_l\}}(\mathcal{X}, \mathcal{X}; \boldsymbol{\theta})$ is width-dependent and weight-dependent. There are two differences between the analytic NTK and the fixed-weight-NTK, but the expression 11 only shows the difference in width and ignores the difference on weights. The underlying reason is that the effects of random weights are offset in an infinite-width network, and thus there is no need to eliminate the randomness by adding an operator such as expectation.

**Another expression of the analytic NTK.** Although not necessary, it is legitimate to add an "expectation" operator in the expression. We rewrite the analytic NTK as the expectation (over random weights) of the limit (over width) of fixed-weight-NTK, expressed mathematically as follows:

$$\Theta(\mathcal{X}, \mathcal{X}) = \mathbb{E}_{\boldsymbol{\theta}} \left[ \lim_{n_l \to \infty, \forall l} \hat{\Theta}^{\{n_l\}}(\mathcal{X}, \mathcal{X}; \boldsymbol{\theta}) \right]. \tag{12}$$

This formula is correct since $\mathbb{E}_{\boldsymbol{\theta}} \left[ \lim_{n_l \to \infty, \forall l} \hat{\Theta}^{\{n_l\}}(\mathcal{X}, \mathcal{X}; \boldsymbol{\theta}) \right] = \mathbb{E}_{\boldsymbol{\theta}} \left[ \Theta(\mathcal{X}, \mathcal{X}) \right] = \Theta(\mathcal{X}, \mathcal{X})$.

**Another approximation of the analytic NTK.** Although the expectation operator is redundant in the Equation 12, it makes a difference when we consider the finite-width approximation. A natural approximation coming out of Equation 12 is the following: first, remove the "lim" operation and use a fixed set of large enough width $\{n_l\}$; second, sample multiple independent weight configurations and use the average as an approximation of the "expectation" $\mathbb{E}_{\boldsymbol{\theta}} \left[ \hat{\Theta}^{\{n_l\}}(\mathcal{X}, \mathcal{X}; \boldsymbol{\theta}) \right]$ for a given set of large enough width $\{n_l\}$. Specifically, with $R$ number of independent initialization points, we obtain the stabilized version of the saliency criterion Equation 8.

### B.2    Understanding multi-sampling formulation through the fixed-weight-NTK evolution

In the previous section, we analyzed why multi-sampling gives a better estimate of the analytic NTK than a single draw of fixed-weight-NTK. This analysis leads to the design of multi-sampling method. However, it may seem unclear why we want to use the analytic NTK as the performance indicator of a finite-width neural network. In this section, we provide an explanation of the multi-sampling method from another perspective.

When the width goes to infinity, the NTK is almost deterministic and remains asymptotically constant throughout training.

Under such circumstances, the analytic NTK is a good measure of the optimization of the large enough network since it measures the expected behavior over random weight configurations and

training trajectory. Let $\boldsymbol{\theta}_t(\boldsymbol{\theta}_0; t)$ denotes the parameter of the network at training step $t$ with initial random weight configuration $\boldsymbol{\theta}_0$, then

$$\boldsymbol{\Theta}(\mathcal{X}, \mathcal{X}) = \mathbb{E}_{\boldsymbol{\theta}_0, t} \left[ \lim_{n_l \to \infty, \forall l} \hat{\boldsymbol{\Theta}}^{\{n_l\}}(\mathcal{X}, \mathcal{X}; \boldsymbol{\theta}_t(\boldsymbol{\theta}_0; t)) \right]. \tag{13}$$

In the finite-width (real-world) setting, the fixed-weight-NTK no longer stays constant and is a function of time. Though we can no longer describe the whole training process with a single deterministic and constant kernel, we can still describe the process with a sequence of NTKs.

For simplicity, we consider a discrete gradient descent update rule. We can characterize the training dynamics of finite-width neural networks with the sequence $\{\hat{\boldsymbol{\Theta}}_0(\mathcal{X}, \mathcal{X}; \boldsymbol{\theta}_0), \hat{\boldsymbol{\Theta}}_1(\mathcal{X}, \mathcal{X}; \boldsymbol{\theta}_1), \cdots, \hat{\boldsymbol{\Theta}}_E(\mathcal{X}, \mathcal{X}; \boldsymbol{\theta}_E)\}$, which contains all the fixed-weight-NTKs at each time step.

In this case, we hope all the elements in the sequence are well-conditioned so the neural network performs well during the training phase. As a result, we need to quantify the influence of removing a mask on the nuclear norm of these fixed-weight-NTKs.

One possible metric is $\frac{1}{E+1} \sum_{t=0}^{E} \|\hat{\boldsymbol{\Theta}}_t(\mathcal{X}, \mathcal{X}; \boldsymbol{\theta}_t)\|_*$, which is the average of the nuclear norm of the fixed-weight-NTKs computed on the parameter sequence $\{\boldsymbol{\theta}_0, \boldsymbol{\theta}_1, \cdots, \boldsymbol{\theta}_E\}$.

We suspect that the parameter space is well explored during the training. The elements in the parameter sequence $\{\boldsymbol{\theta}_0, \boldsymbol{\theta}_1, \cdots, \boldsymbol{\theta}_E\}$ may be well-spread, and thus $\frac{1}{E+1} \sum_{t=0}^{E} \|\hat{\boldsymbol{\Theta}}_t(\mathcal{X}, \mathcal{X}; \boldsymbol{\theta}_t)\|_*$ can be viewed as an approximation of $\mathbb{E}_{\boldsymbol{\theta}} \left[ \|\hat{\boldsymbol{\Theta}}(\mathcal{X}, \mathcal{X}; \boldsymbol{\theta})\|_* \right]$. As another way to approximate this expectation, we can approximate $\mathbb{E}_{\boldsymbol{\theta}} \left[ \|\hat{\boldsymbol{\Theta}}(\mathcal{X}, \mathcal{X}; \boldsymbol{\theta})\|_* \right]$ by multiple random weight configurations as $\sum_{r=1}^{R} \|\hat{\boldsymbol{\Theta}}_{0,r}(\mathcal{X}, \mathcal{X}; \boldsymbol{\theta}_{0,r})\|_*$, which is the saliency criterion Equation 8.

As a comparison, many existing works compute the fixed-weight-NTK with a single random weight configuration at initialization and are hence not a very good measure of the optimization of the network with practical size. In our work, we compute the expected fixed-weight-NTK and we believe such an expected version of the fixed-weight-NTK better captures the expected behavior of the subnetworks over random weight configurations and training trajectory. Hence, we aim to find a subnetwork with a well-conditioned expected fixed-weight-NTK.

To make the expected fixed-weight-NTK well-conditioned, we choose to align it to the fixed-weight-NTK of the dense network at initialization. Such a choice is justified for the following two reasons:

- **The eigenspectrum of the dense fixed-weight-NTK is a good candidate.** We believe that the good performance of the dense network implies that the eigenspectrum of the dense network at initialization is well-conditioned (See Section 6.5 and Appendix G.1 for the eigenspectrum of the dense network.).

- **The eigenspectrum of the dense fixed-weight-NTK is a practical choice.** For foresight pruning, we can only access the fixed-weight-NTK at initialization. Also, the foresight pruning framework mentioned in Section 3 is a general framework that preserves a scalar indicator of the dense network. Hence, due to the intrinsic nature of the framework, we can only try to align the eigenspectrum of the sparse network to the dense counterpart at initialization, rather than some given well-conditioned spectrum.

Indeed, for practical networks, the eigenspectrum of the dense network may be changing during the training procedure. Hence, our method can only approximately align the training behavior to the dense network at the early training stage. For the later epochs, the expected well-conditioned property of the subnetwork ensures that the determined subnetworks have a better optimization performance compared to other foresight pruning methods.

## C  ABLATION STUDIES ON THE NUMBER OF ITERATIVE STEPS

We show experiment results in Section 6.4 here. Figure 3 shows that for smaller datasets, we can use a smaller number of iterations $T$ for NTK-SAP without seeing a significant performance drop.

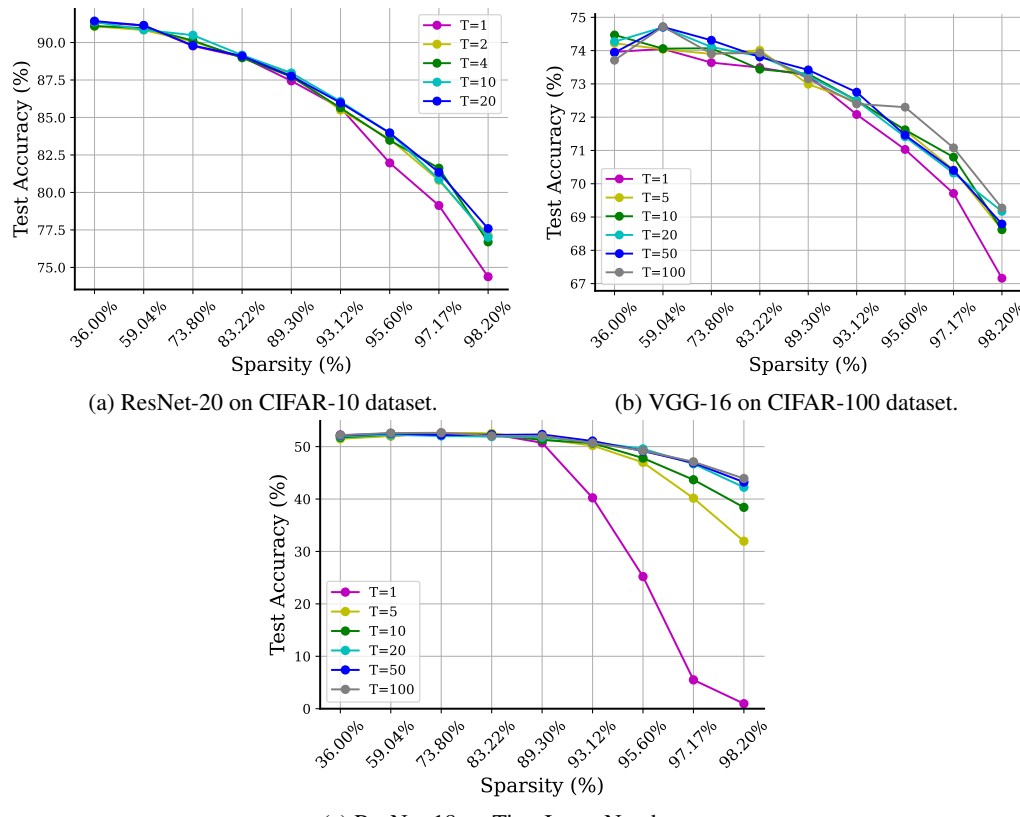

(a) ResNet-20 on CIFAR-10 dataset.

(b) VGG-16 on CIFAR-100 dataset.

(c) ResNet-18 on Tiny-ImageNet dataset.

Figure 3: Evaluate the effect of number of iterations ($T$) on performance of NTK-SAP on CIFAR-10 (ResNet-20), CIFAR-100 (VGG-16), and Tiny-ImageNet (ResNet-18).

## D DETAILED COMPUTATIONAL COST

### D.1 DETAILED COMPUTATIONAL TIME COMPARISON

Table 3: Pruning time of considered methods compared to time for 1 epoch of training.

| Dataset | CIFAR-10 | CIFAR-100 | Tiny-ImageNet | ImageNet | |
|---|---|---|---|---|---|
| Model | ResNet-20 | VGG-16 | ResNet-18 | ResNet-18 | ResNet-50 |
| Training time of 1 epoch | 9.6 s | 11 s | 24 s | 15 min 42 s | 18 min |
| SNIP | 1 s | 1 s | 1 s | 13 s | 34 s |
| GraSP | 1 s | 1 s | 3 s | 50 s | 5 min 32 s |
| Synflow | 12 s | 34 s | 1 min 2 s | 6 min 34 s | 5 min 39 s |
| Iterative SNIP | 24 s | 46 s | 1 min 18 s | 16 min 42 s | 23 min 17 s |
| NTK-SAP | 17 s | 1 min 14 s | 2 min 14 s | 17 min 53 s | 22 min 49 s |
| NTK-SAP-S | 3 s | 14 s | 31 s | - | - |

We include a detailed comparison of the pruning time of considered methods along with the time for 1 epoch of model training in Table 3. We are interested in comparing the pruning time of NTK-SAP with the training time of 1 epoch since it is empirically shown by de Jorge et al. (2020b) that most foresight pruning methods are not worse than magnitude pruning after 1 round of training. We also repeat such experiments in Appendix E.

For ImageNet experiments, the pruning time of NTK-SAP is similar to the time for 1-2 epochs of model training. As for smaller datasets, Appendix C shows that we can use a smaller number of pruning rounds $T$. Thus we instead compare the rest of the methods to NTK-SAP with a smaller $T$ for smaller datasets. We name such a modified version `NTK-SAP-S`. Specifically, we use $T = 4$ for CIFAR-10 experiment and use $T = 20$ for CIFAR-100 and Tiny-ImageNet. We can see that `NTK-SAP-S` is able to finish pruning in 1-2 epochs of training time. Thus, we believe that the computational cost of NTK-SAP is reasonable.

## D.2 DETAILED FLOPs COMPARISON

In this section, we compare different foresight pruning methods in terms of FLOPs. We use the default FLOPs counter in the Synflow codebase. We approximate backward FLOPs twice as the inference/forward FLOPs. Pruning FLOPs are calculated with a simplification that the neural network is dense when computing the gradients.

**Pruning cost.** Comparison of pruning cost for different foresight pruning methods are present in Table 4. Synflow has the lowest pruning cost since its saliency score is computed with input containing all 1s. NTK-SAP uses roughly twice the pruning cost compared to Iterative SNIP, making it the method which requires more costs compared to other foresight pruning methods. However, we want to emphasize that the pruning cost of NTK-SAP can be reduced by using smaller $T$ as shown in Section 6.4 and Section C.

**Inference cost.** We further compare the inference cost of subnetworks found by different foresight pruning methods. Results are present in Table 5, 6 and 7.

**Total pruning and training cost.** Finally, we compare the total pruning and training FLOPs of considered foresight pruning methods in Table 8, 9 and 10. As discussed in Section 6.4, NTK-SAP generally needs more total computational costs. However, we believe that the additional computational cost is reasonable, especially considering that NTK-SAP is a data-agnostic method and can be easily applied to different datasets once the mask is found. Furthermore, we again want to emphasize the possibility of decreasing the computational cost of NTK-SAP by reducing $T$.

## E EARLY PRUNING BASELINE

We repeat experiments from FORCE (de Jorge et al., 2020b) where we compare mentioned foresight pruning methods against early pruning. Precisely, we train the dense neural network for 1 epoch and then prune the trained neural network using magnitude pruning. Then the weights are rewound to their initial values as in the LTH work. We term such baseline as `Mag-1epoch`. Figure 4 illustrates that 1 epoch of training slightly improves the performance of magnitude pruning. However, NTK-SAP still outperforms such an early pruning method.

## F SNIP VARIANTS

In this section, we investigate variants of SNIP, including SNIP, Iterative SNIP, and FORCE.

**Effects of different sparsity schedules.** We first investigate the effects of the sparsity schedule used in Iterative SNIP. Specifically, we compare Iterative SNIP (Exp), which is used in the main paper, with Iterative SNIP (Linear), where we use a linear sparsity schedule. As is shown in Figure 5, we find that a linear schedule slightly improves the performance of Iterative SNIP on smaller datasets. However, Table 11 shows that the linear schedule does not necessarily improve the performance of Iterative SNIP on larger datasets.

The performance gap between Iterative SNIP implemented in this work and what is reported in (de Jorge et al., 2020b) may be due to the different initialization methods and the number of batches used for pruning. Additionally, de Jorge et al. (2020b) only considers relatively large neural networks, including VGG-19 and ResNet-50. However, note that it is also empirically shown by Frankle et al.

Table 4: Pruning FLOPs comparison. Results with the largest FLOPs are in blue and results with the smallest FLOPs are in **bold**.

| Dataset | CIFAR-10 | CIFAR-100 | Tiny-ImageNet |
|---|---|---|---|
| Model | ResNet-20 | VGG-16 | ResNet-18 |
| SNIP | $2.48 \times 10^{11}$ | $1.88 \times 10^{12}$ | $8.94 \times 10^{11}$ |
| GraSP | $4.95 \times 10^{11}$ | $3.77 \times 10^{12}$ | $1.79 \times 10^{12}$ |
| Synflow | $\mathbf{1.24 \times 10^{10}}$ | $\mathbf{9.42 \times 10^{10}}$ | $\mathbf{4.47 \times 10^{10}}$ |
| Iterative SNIP | $2.48 \times 10^{13}$ | $1.88 \times 10^{14}$ | $8.94 \times 10^{13}$ |
| NTK-SAP | $4.95 \times 10^{13}$ | $3.77 \times 10^{14}$ | $1.79 \times 10^{14}$ |
| NTK-SAP (small $T$) | $9.91 \times 10^{12}$ | $7.54 \times 10^{13}$ | $3.58 \times 10^{13}$ |

Table 5: Inference FLOPs ($\times 10^6$) comparison on CIFAR-10(ResNet-20). Results with the largest FLOPs are in blue and results with the smallest FLOPs are in **bold**.

| Sparsity | 36.00% | 73.80% | 89.30% | 95.60% | 98.20% |
|---|---|---|---|---|---|
| SNIP | 31.90 | 19.40 | 12.30 | 7.37 | 3.77 |
| GraSP | **24.20** | **13.50** | 8.65 | 4.95 | 3.20 |
| Synflow | 33.20 | 21.60 | 12.20 | 5.73 | 2.67 |
| Iterative SNIP | 31.30 | 16.70 | **7.80** | **3.46** | **2.10** |
| NTK-SAP | 31.30 | 18.20 | 10.10 | 5.30 | 2.83 |

Table 6: Inference FLOPs ($\times 10^7$) comparison on CIFAR-100(VGG-16). Results with the largest FLOPs are in blue and results with the smallest FLOPs are in **bold**.

| Sparsity | 36.00% | 73.80% | 89.30% | 95.60% | 98.20% |
|---|---|---|---|---|---|
| SNIP | 25.50 | 17.10 | 11.80 | 8.19 | 5.34 |
| GraSP | **18.30** | **10.30** | **6.15** | **3.75** | **2.48** |
| Synflow | 25.80 | 17.60 | 11.50 | 6.95 | 3.71 |
| Iterative SNIP | 25.20 | 15.80 | 9.67 | 5.51 | 2.97 |
| NTK-SAP | 26.10 | 17.50 | 11.60 | 7.54 | 4.58 |

Table 7: Inference FLOPs ($\times 10^7$) comparison on Tiny-ImageNet(ResNet-18). Results with the largest FLOPs are in blue and results with the smallest FLOPs are in **bold**.

| Sparsity | 36.00% | 73.80% | 89.30% | 95.60% | 98.20% |
|---|---|---|---|---|---|
| SNIP | 12.30 | 8.62 | 6.13 | 4.14 | 2.51 |
| GraSP | **8.14** | **5.29** | **3.59** | 2.45 | 1.65 |
| Synflow | 12.80 | 9.49 | 6.77 | 4.31 | 2.36 |
| Iterative SNIP | 12.20 | 7.91 | 4.40 | **2.00** | **0.80** |
| NTK-SAP | 12.70 | 9.47 | 6.77 | 4.53 | 2.73 |

Table 8: Total pruning + training FLOPs ($\times 10^{14}$) comparison on CIFAR-10(ResNet-20). Results with the largest FLOPs are in blue and results with the smallest FLOPs are in **bold**.

| Sparsity | 36.00% | 73.80% | 89.30% | 95.60% | 98.20% |
|---|---|---|---|---|---|
| SNIP | 7.65 | 4.67 | 2.95 | 1.77 | 0.91 |
| GraSP | **5.81** | **3.24** | **2.08** | 1.19 | 0.77 |
| Synflow | 7.98 | 5.19 | 2.93 | 1.38 | **0.64** |
| Iterative SNIP | 7.75 | 4.25 | 2.12 | **1.08** | 0.75 |
| NTK-SAP | 8.01 | 4.86 | 2.91 | 1.77 | 1.17 |

Table 9: Total pruning + training FLOPs ($\times 10^{15}$) comparison on CIFAR-100(VGG-16). Results with the largest FLOPs are in blue and results with the smallest FLOPs are in **bold**.

| Sparsity | 36.00% | 73.80% | 89.30% | 95.60% | 98.20% |
|---|---|---|---|---|---|
| SNIP | 6.13 | 4.09 | 2.83 | 1.97 | 1.28 |
| GraSP | **4.39** | **2.47** | **1.48** | **0.90** | **0.60** |
| Synflow | 6.19 | 4.23 | 2.76 | 1.67 | 0.89 |
| Iterative SNIP | 6.24 | 3.99 | 2.51 | 1.51 | 0.90 |
| NTK-SAP | 6.65 | 4.58 | 3.17 | 2.19 | 1.48 |

Table 10: Total pruning + training FLOPs ($\times 10^{14}$) comparison on Tiny-ImageNet(ResNet-18). Results with the largest FLOPs are in blue and results with the smallest FLOPs are in **bold**.

| Sparsity | 36.00% | 73.80% | 89.30% | 95.60% | 98.20% |
|---|---|---|---|---|---|
| SNIP | 73.70 | 51.70 | 36.80 | 24.80 | 15.10 |
| GraSP | **48.80** | **31.80** | **21.50** | 14.70 | 9.91 |
| Synflow | 76.70 | 57.00 | 40.60 | 25.90 | 14.10 |
| Iterative SNIP | 73.90 | 48.40 | 27.30 | **12.90** | **5.71** |
| NTK-SAP | 78.20 | 58.60 | 42.40 | 29.00 | 18.10 |

Table 11: Comparison of SNIP variants on ImageNet dataset. Best results are in bold.

| Network | **ResNet-18** | | **ResNet-50** | |
|---|---|---|---|---|
| Sparsity percentage | 89.26% | 95.60% | 89.26% | 95.60% |
| (Dense Baseline) | 69.98 | | 76.20 | |
| SNIP Lee et al. (2018) | **57.41** | **45.04** | **60.98** | **40.69** |
| Iterative SNIP (Exp) de Jorge et al. (2020b) | 52.97 | 37.44 | 52.53 | 36.82 |
| Iterative SNIP (Linear) | 50.29 | 40.98 | 56.54 | 30.55 |

(2020b) and Tanaka et al. (2020)[9] that Iterative SNIP does not necessarily improve the performance of SNIP.

**Unstable performance of FORCE.** In addition, FORCE (de Jorge et al., 2020b), which allows pruned masks to resurrect in the process of iterative pruning, is also considered. Results are also presented in Figure 5. We observe a significant performance drop of FORCE on VGG-16 (CIFAR-100). As a result, we do not report FORCE in the main paper.

---

[9]See the author feedback file of the Synflow paper.

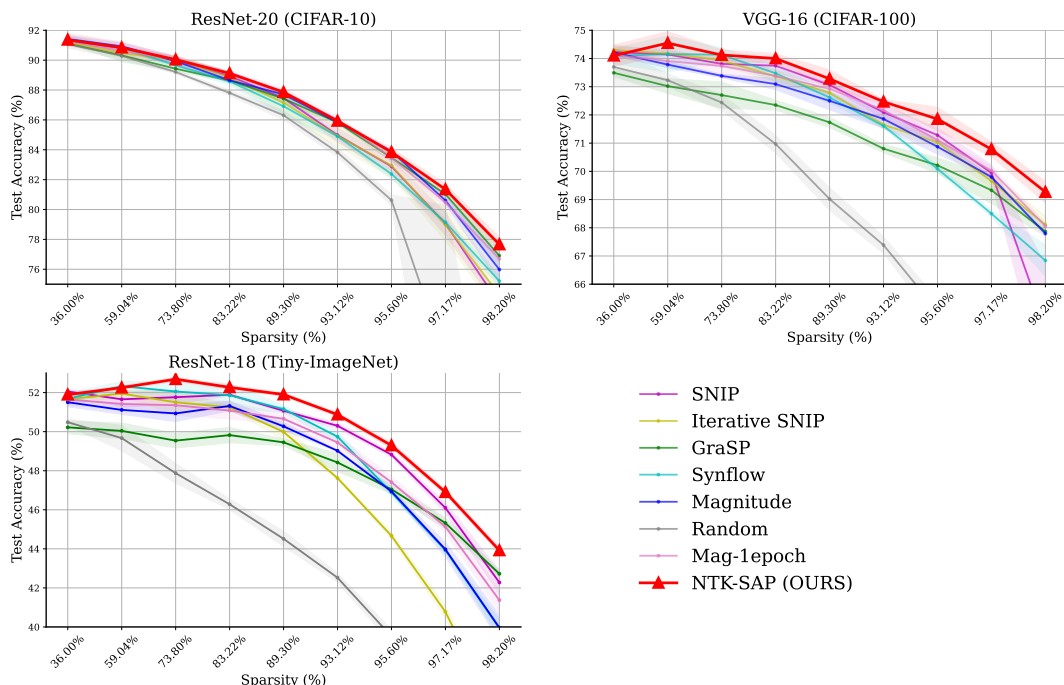

Figure 4: Comparison of foresight pruning methods against magnitude pruning after 1 epoch of training. Results are averaged over 3 random seeds and the shaded areas denote the standard deviation of the runs.

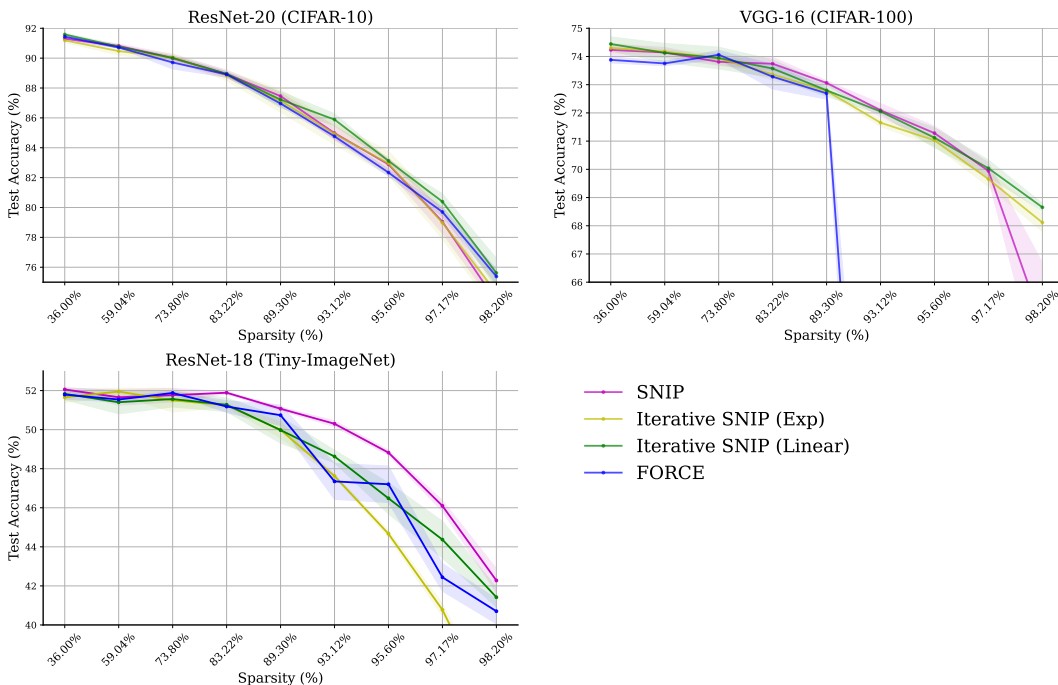

Figure 5: Comparison of SNIP variants. Results are averaged over 3 random seeds and the shaded areas denote the standard deviation of the runs.

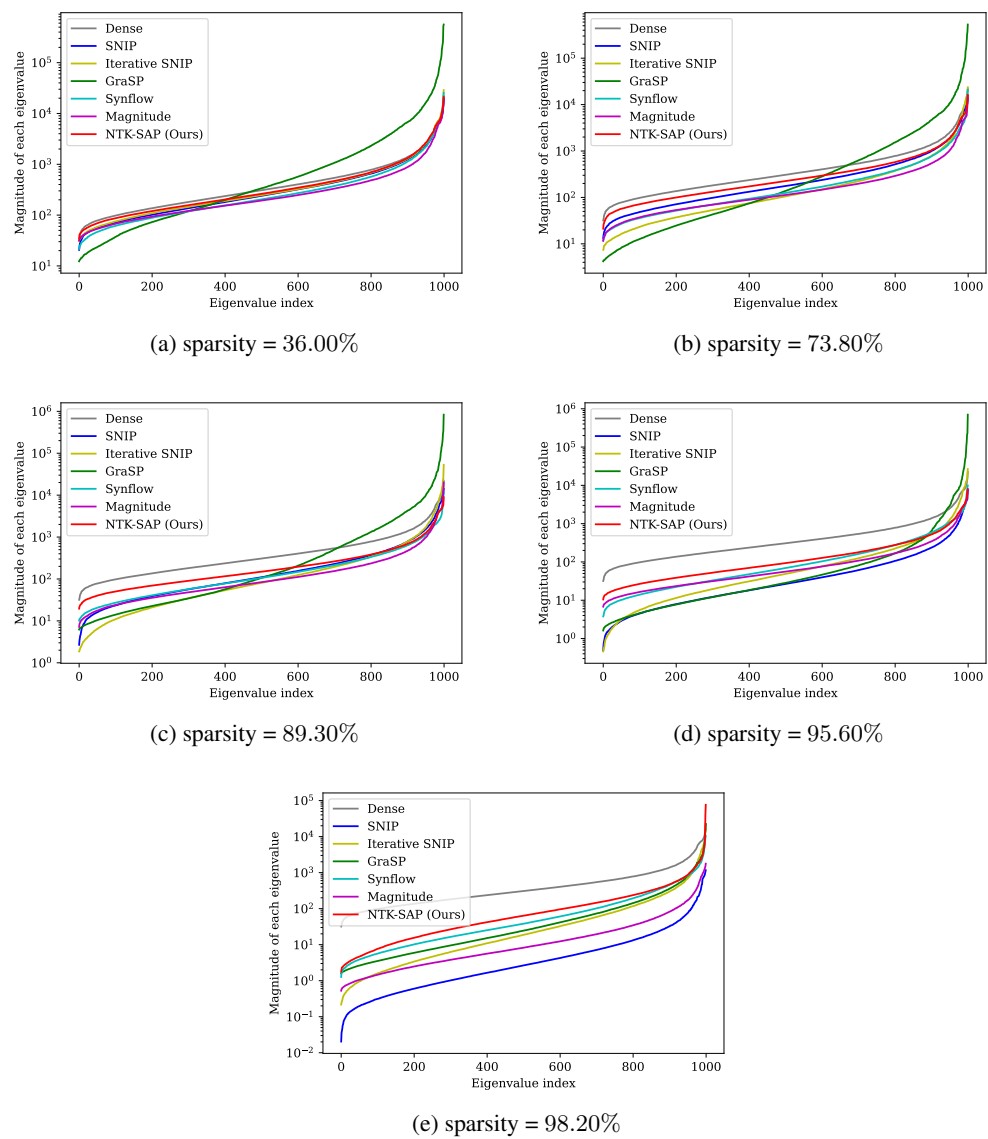

Figure 6: Eigenvalue distribution of the fixed-weight-NTK on ResNet-20 (CIFAR-10) at different sparsity ratios. We put the index of each eigenvalue on the x-axis, and the corresponding eigenvalue magnitude is presented on the y-axis.

## G  EIGENSPECTRUM OF PRUNED NEURAL NETWORKS

### G.1  EIGENSPECTRUM AT INITIALIZATION

To visualize the effectiveness of our proposed NTK-SAP method, we conduct experiments on CIFAR-10 dataset with ResNet-20 at sparsity ratios $\{36.00\%, 73.80\%, 89.30\%, 95.60\%, 98.20\%\}$.[10]  We compute the fixed-weight-NTK with a batch size of 100. Thus the fixed-weight-NTK is of size $1000 \times 1000$.

---

[10]Other datasets have a much larger value of $k$, and thus the fixed-weight-NTK is much more difficult to compute.

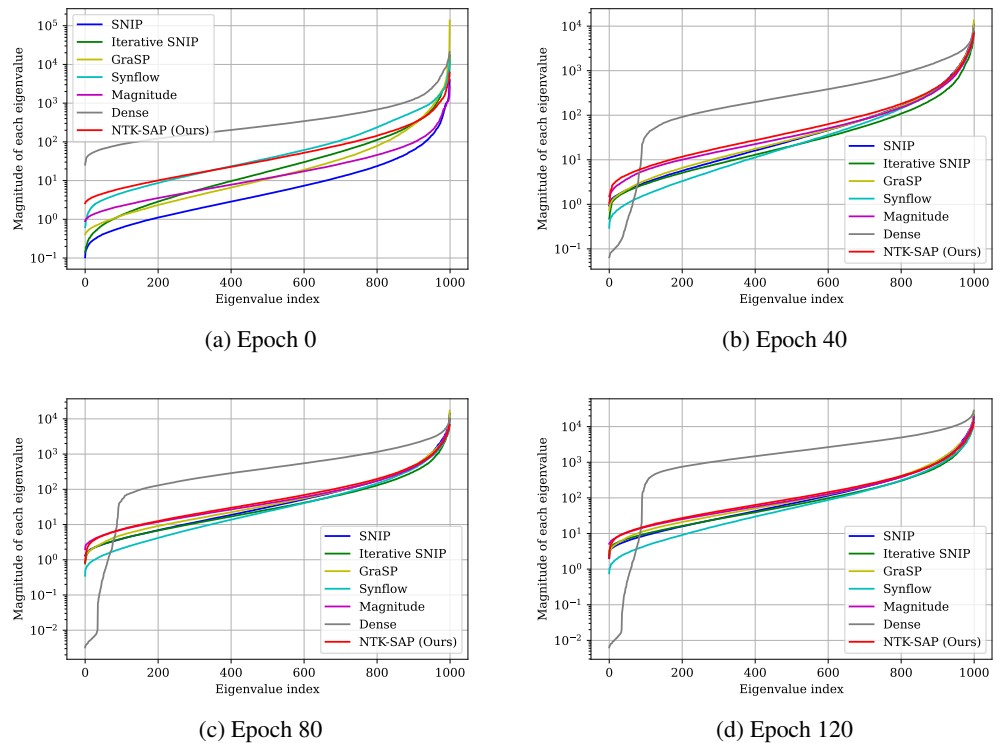

(a) Epoch 0                                 (b) Epoch 40

(c) Epoch 80                              (d) Epoch 120

Figure 7: Eigenvalue distribution of the fixed-weight-NTK on ResNet-20 (CIFAR-10) with different training epochs at sparsity ratio 98.20%. We put the index of each eigenvalue on the x-axis, and the corresponding eigenvalue magnitude is presented on the y-axis.

Figure 6 shows eigenvalue distributions of neural networks pruned by mentioned foresight pruning methods. It illustrates the effectiveness of NTK-SAP in keeping the spectrum close to the dense network.

As is shown in Figure 7, we see that the spectrum of the dense network is changing throughout the training. On the contrary, the fixed-weight-NTK eigenspectrum of all the pruned networks shows similar behavior and deviates from the dense one. However, we show that, compared to other foresight pruning methods, the eigenspectrum of NTK-SAP aligned with the dense counterpart for the first few epochs (As it is shown that the majority of the eigenvalues of NTK-SAP are the closest to the dense network for most of the time.). Secondly, the fixed-weight-NTK of NTK-SAP is well-conditioned throughout the entire training process, which aligns with our motivation.

### G.2 EIGENSPECTRUM DURING TRAINING

We also plot eigenvalue distributions of neural networks pruned by these foresight pruning methods during training. Since the test performance of different methods does not vary too much on the CIFAR-10 dataset for a wide range of sparsity ratios, we only focus on the most extreme sparsity, i.e, 98.20%. We report relative eigenvalues by dividing each eigenvalue by the average of 10% smallest eigenvalues. Results are presented in Figure 8.

We observe that the best three methods on the CIFAR-10 dataset, i.e, NTK-SAP, GraSP, and Magnitude, have relatively low condition numbers. Furthermore, the best two methods, i.e., GraSP and NTK-SAP, have almost stable/unchanging relative eigenvalue distribution during training. That could possibly explain why GraSP and NTK-SAP perform well at this sparsity ratio.

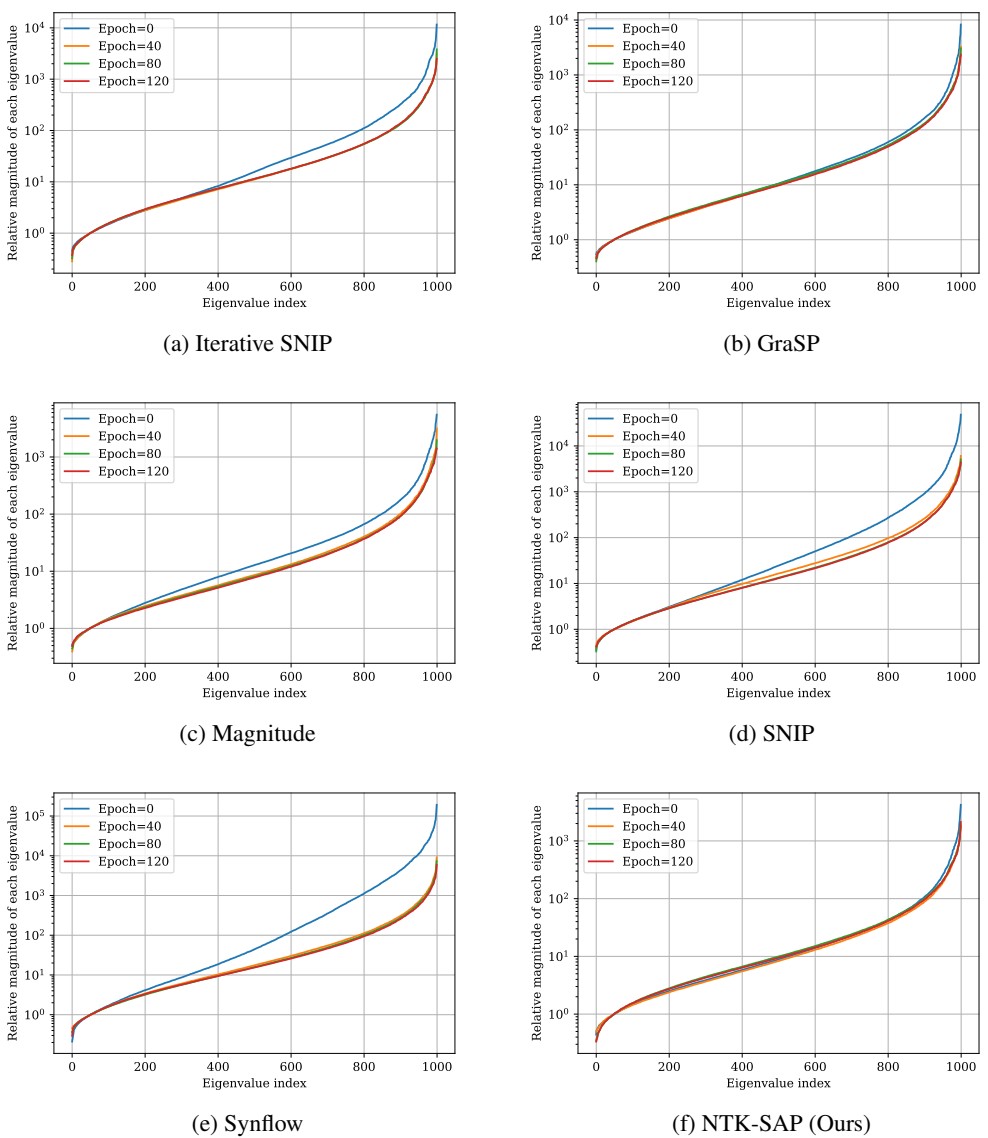

Figure 8: Relative eigenvalue distribution of the fixed-weight-NTK during training on ResNet-20 (CIFAR-10) at sparsity 98.20%. We put the index of each eigenvalue on the x-axis, and the corresponding relative eigenvalue magnitude is presented on the y-axis.

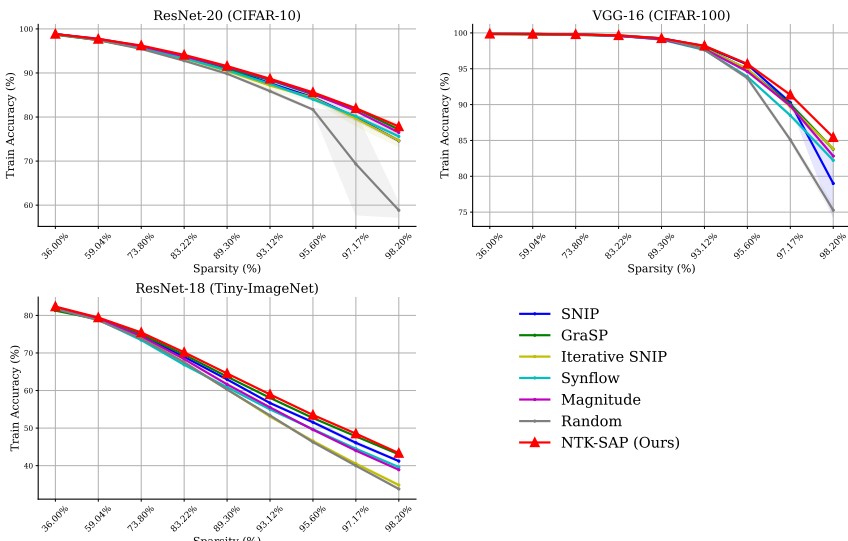

Figure 9: Train accuracy of NTK-SAP against other foresight pruning methods. Results are averaged over 3 random seeds and the shaded areas denote the standard deviation of the runs.

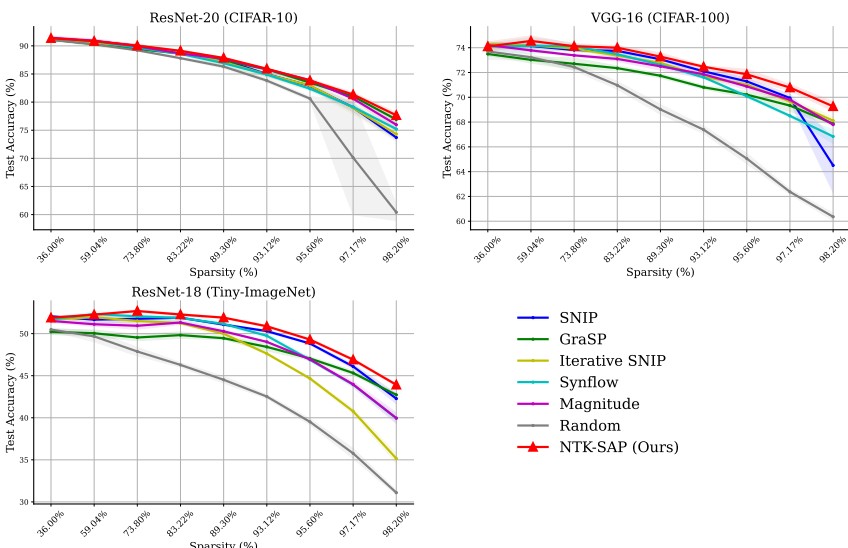

Figure 10: Test accuracy of NTK-SAP against other foresight pruning methods. Results are averaged over 3 random seeds and the shaded areas denote the standard deviation of the runs.

Table 12: Ablation study of perturbation hyper-parameter $\epsilon$ on CIFAR-10 (ResNet-20). Best results are in **bold**.

| Sparsity | 36.00% | 73.80% | 89.30% | 95.60% | 98.20% |
|---|---|---|---|---|---|
| $\epsilon = 1.0 \times 10^{-4\dagger}$ | $91.36 \pm 0.41$ | $90.04 \pm 0.23$ | $87.86 \pm 0.13$ | $83.84 \pm 0.28$ | $\mathbf{77.66 \pm 0.50}$ |
| $\epsilon = 2.5 \times 10^{-5}$ | $91.34 \pm 0.11$ | $90.30 \pm 0.17$ | $87.94 \pm 0.20$ | $83.82 \pm 0.24$ | $77.33 \pm 0.10$ |
| $\epsilon = 1.0 \times 10^{-5}$ | $\mathbf{91.45 \pm 0.13}$ | $90.26 \pm 0.06$ | $87.64 \pm 0.18$ | $\mathbf{83.92 \pm 0.44}$ | $77.53 \pm 0.34$ |
| $\epsilon = 1.0 \times 10^{-6}$ | $91.41 \pm 0.03$ | $\mathbf{90.44 \pm 0.33}$ | $\mathbf{88.00 \pm 0.36}$ | $83.66 \pm 0.23$ | $76.90 \pm 0.44$ |

† Used in the experiments of the main body.

Table 13: Ablation study of perturbation hyper-parameter $\epsilon$ on CIFAR-100 (VGG-16). Best results are in **bold**.

| Sparsity | 36.00% | 73.80% | 89.30% | 95.60% | 98.20% |
|---|---|---|---|---|---|
| $\epsilon = 1.0 \times 10^{-4\dagger}$ | $74.10 \pm 0.37$ | $\mathbf{74.12 \pm 0.19}$ | $73.28 \pm 0.32$ | $\mathbf{71.85 \pm 0.41}$ | $\mathbf{69.27 \pm 0.39}$ |
| $\epsilon = 2.5 \times 10^{-5}$ | $74.25 \pm 0.16$ | $74.09 \pm 0.20$ | $73.20 \pm 0.24$ | $71.49 \pm 0.08$ | $68.97 \pm 0.16$ |
| $\epsilon = 1.0 \times 10^{-5}$ | $\mathbf{74.31 \pm 0.41}$ | $74.08 \pm 0.20$ | $73.18 \pm 0.39$ | $71.68 \pm 0.25$ | $69.24 \pm 0.21$ |
| $\epsilon = 1.0 \times 10^{-6}$ | $74.27 \pm 0.22$ | $73.90 \pm 0.25$ | $\mathbf{73.37 \pm 0.37}$ | $71.49 \pm 0.13$ | $69.26 \pm 0.48$ |

† Used in the experiments of the main body.

Table 14: Ablation study of perturbation hyper-parameter $\epsilon$ on Tiny-ImageNet (ResNet-18). Best results are in **bold**.

| Sparsity | 36.00% | 73.80% | 89.30% | 95.60% | 98.20% |
|---|---|---|---|---|---|
| $\epsilon = 1.0 \times 10^{-4\dagger}$ | $51.89 \pm 0.38$ | $\mathbf{52.69 \pm 0.14}$ | $51.90 \pm 0.12$ | $49.29 \pm 0.15$ | $\mathbf{43.92 \pm 0.39}$ |
| $\epsilon = 2.5 \times 10^{-5}$ | $\mathbf{52.10 \pm 0.51}$ | $52.18 \pm 0.16$ | $51.95 \pm 0.26$ | $\mathbf{49.48 \pm 0.38}$ | $43.63 \pm 0.19$ |
| $\epsilon = 1.0 \times 10^{-5}$ | $51.86 \pm 0.16$ | $51.97 \pm 0.46$ | $\mathbf{52.00 \pm 0.30}$ | $48.83 \pm 0.09$ | $43.36 \pm 0.53$ |
| $\epsilon = 1.0 \times 10^{-6}$ | $51.94 \pm 0.45$ | $52.31 \pm 0.19$ | $51.32 \pm 0.20$ | $49.15 \pm 0.38$ | $42.94 \pm 0.22$ |

† Used in the experiments of the main body.

## H    FULL PLOTS OF THE MAIN EXPERIMENTS

We further include the complete plots of train/test accuracy of the main experiments reported in Section 6.1. Results are shown in Figure 9 and Figure 10. NTK-SAP provides higher training accuracy than other foresight pruning methods, showing its capability to identify subnetworks that are easier to train.

Generally, the test performance of all foresight pruning methods degrades with more sparsity induced. And the speed of performance degradation accelerates as the sparsity ratio increases. Additionally, for VGG-16 (CIFAR-100) and ResNet-18 (Tiny-ImageNet), it can be clearly seen that the test performance of NTK-SAP first increases and then decreases, which shows the generalization improvements brought by pruning. However, it is not observed in ResNet-20 (CIFAR-10).

## I    ABLATION STUDY ON PERTURBATION HYPER-PARAMETER

In this section, we conduct an ablation study on the effects of choosing different values of perturbation hyper-parameter $\epsilon$. We choose $\epsilon$ from $\{1.0 \times 10^{-4}, 2.5 \times 10^{-5}, 1.0 \times 10^{-5}, 1.0 \times 10^{-6}\}$, where $1.0 \times 10^{-4}$ is the value we used in the main body. Results on CIFAR-10, CIFAR-100, and Tiny-ImageNet are shown in Table 12, 13, and 14, respectively.

The tables show that NTK-SAP is robust to the choices of perturbation hyper-parameter $\epsilon$.

Table 15: Overview of considered foresight pruning methods.

| Pruning method | Weight agnostic | Data-free | Iterative method |
|---|---|---|---|
| SNIP (Lee et al., 2018) | | | |
| GraSP (Wang et al., 2020) | | | |
| ProsPr (Alizadeh et al., 2022) | | | |
| FORCE (de Jorge et al., 2020b) | | | ✓ |
| Iterative SNIP (de Jorge et al., 2020b) | | | ✓ |
| Synflow (Tanaka et al., 2020) | | ✓ | ✓ |
| NTK-SAP (Ours) | ✓ | ✓ | ✓ |

## J  OVERVIEW OF THE CONSIDERED FORESIGHT PRUNING METHODS

**Synflow and NTK-SAP are the only two data-free foresight pruning methods.**  In this section, we summarize all the mentioned foresight pruning methods in Table 15. Synflow and NTK-SAP are the only foresight pruning methods that do not need any data. It indicates that once pruning is done, subnetworks found can be easily applied to different datasets. Moreover, to our best knowledge, NTK-SAP is the first foresight pruning method that is weight agnostic.

## K  DIFFERENCE BETWEEN NTK-SAP AND NEURAL TANGENT TRANSFER

In this section, we briefly discuss the difference between NTK-SAP and Neural Tangent Transfer (NTT) (Liu & Zenke, 2020). NTT also uses the theory of NTK to guide foresight pruning. Its goal is to jointly optimize $\mathbf{m}$ and $\boldsymbol{\theta}$ so that the output and the fixed-weight-NTK of the pruned model are as close as to the dense counterpart.

However, the computational cost of explicitly constructing the fixed-weight-NTK is much larger than that of estimating the spectrum used in NTK-SAP. As a result, NNT may be too expensive to use for large models including ResNet-18, ResNet-34, and ResNet-50.[11] Indeed, the original NTT paper only considers small networks, including multi-layer perceptrons, LeNet, and Conv-4.

Another question is: can we apply the approximation used in NTK-SAP to NTT? We believe it is not trivial. The idea of NTK-SAP is to remove those masks that have the least influence on the fixed-weight-NTK spectrum. Hence, the spectrum is likely to stay roughly the same after pruning. On the contrary, NTT, as an optimization-based method, may need extra efforts to modify the loss to penalize the change of the fixed-weight-NTK spectrum since two spectrums can have the exact summation but with totally different cumulative distribution functions (CDF).

## L  COMPARISON WITH PROSPR

Before submission, it has come to our attention that ProsPr (Alizadeh et al., 2022) has been published recently as a new competitive foresight pruning method. We tried to reproduce the results of ProsPr so that we can conduct a fair comparison between ProPr and NTK-SAP.[12] Nevertheless, we are unable to reproduce the reported results in the original ProsPr paper. We suspect that this is because we did not use the same hyperparameters as the authors, and then asked the authors for their hyperparameters. The authors told us that the hyper-parameters information is not yet released and will be released later some time. We will conduct such a comparison once this information is available.

---

[11] See related discussions in issue #2 of NTT GitHub page.

[12] Note that there are some differences between architectures used in ProsPr and the benchmark they used (Frankle et al., 2020b).

However, we want to emphasize that, ProsPr, as a data-dependent method (See Table 15), requires data to perform the pruning procedure. On the contrary, NTK-SAP does not require any data. Once the model is pruned, it can be applied to any datasets without additional computation.

## M   DISCUSSION ON THE WEIGHT-AGNOSTIC PROPERTY OF NTK-SAP

In Section 5.3 we have discussed the weight-agnostic property of NTK-SAP. Here we include a discussion on if a good pruning solution should also be weight-agnostic, at least in the sense of expectation.

We think it depends on the type of pruning methods. For post-hoc pruning, a good pruning solution may be weight-dependent. For example, when using one good post-hoc pruning method "lottery ticket algorithm" (Frankle & Carbin, 2018), mask-dependent weight configuration performs better than random weight configurations.

However, as pointed out by (Frankle et al., 2020b), most existing foresight pruning methods are not very sensitive to newly sampled initialization. One reason could be that foresight pruning methods do not have access to enough information to learn the local features/patterns of the images (Pellegrini & Biroli, 2022) or the basin of the pruning solution (Evci et al., 2022). So in this sense, a good solution for foresight pruning may be weight-agnostic. Anyhow, this argument is not decisive evidence that the best foresight pruning method shall be weight-agnostic.

Another question may be since NTK-SAP intends to find subnetworks that perform well in expectation, what happens if a really bad initialization that has zero overlaps with the binary mask? Will this network be trained well? We believe that such a network indeed should not be trained well. However, the probability of such a situation is 0 since known initialization methods, e.g., Xavier or Kaiming normalization, draw initial weights from continuous random distributions. Furthermore, if a worst-case initialization is allowed, a dense network can also encounter an all-zero initialization which leads to poor results.

## N   WHY NTK-SAP CAN POTENTIALLY AVOID LAYER COLLAPSE

In Section 5.3 we mention that NTK-SAP uses iterative pruning to potentially avoid layer collapse. We would like to give a possible explanation of why NTK-SAP implicitly avoids layer collapse for the following reasons:

Firstly, Synflow (Tanaka et al., 2020) proves that synaptic saliency-based foresight pruning methods broadly follow conservation law and iterative pruning can help them avoid layer collapse. As a result, NTK-SAP, as an iterative synaptic saliency-based pruning method, could also theoretically avoid layer collapse.

Secondly, Lee et al. (2019b) is a re-initialization method based on SNIP aiming at improving signal propagation when layer collapse is about to happen. This work reveals that when layer collapse is about to occur, the spectrum of the layer-wise input-output Jacobian is extremely poor so it breaks layer-wise dynamical isometry. Similarly, the spectrum of the NTK can also serve as such an indicator. Specifically, it can be proved that when layer collapse is about to happen, the spectrum of the NTK deteriorates as well. Here, we would like to give a simple example for a better understanding. For example, when layer collapse happens, the weight gradients of DNNs will be zeros and the fix-weight-NTK will be a zero matrix with all zero eigenvalues. In this sense, NTK-SAP works in a way very similar to Iterative SNIP (de Jorge et al., 2020b) to avoid layer collapse.

## O  JUSTIFICATION OF FINITE DIFFERENCE FORMULATION

In this section we justify the finite difference formulation we used in Section 5.2. To justify this approximation, we have

$$
\begin{aligned}
&\mathbb{E}_{\Delta\boldsymbol{\theta}\sim\mathcal{N}(\mathbf{0},\epsilon\mathbf{I})}\left[\|f(\mathcal{X};\boldsymbol{\theta}_0) - f(\mathcal{X};\boldsymbol{\theta}_0 + \Delta\boldsymbol{\theta})\|_2^2\right] \\
\approx&\mathbb{E}_{\Delta\boldsymbol{\theta}}\left[\|\nabla_{\boldsymbol{\theta}_0}f(\mathcal{X};\boldsymbol{\theta}_0)\Delta\boldsymbol{\theta}\|_2^2\right] \\
=&\sum_{i,j}[\nabla_{\boldsymbol{\theta}_0}f(\mathcal{X};\boldsymbol{\theta}_0)]_{i,j}^2\,\mathbb{E}\left[\Delta\theta_j\Delta\theta_j\right] \\
=&\epsilon\|\nabla_{\boldsymbol{\theta}_0}f(\mathcal{X};\boldsymbol{\theta}_0)\|_F^2.
\end{aligned}
$$

