# OpenReview forum: "NTK-SAP: Improving neural network pruning by aligning training dynamics"
_ICLR.cc/2023/Conference — ICLR 2023 poster_

### Official Review · Reviewer_8ZBV · 2022-10-22

**Confidence:** 3
**Correctness:** 3
**Technical Novelty And Significance:** 3
**Empirical Novelty And Significance:** Not applicable
**Recommendation:** 6

**Clarity, Quality, Novelty And Reproducibility:**

The paper is clear and easy to follow. The method is novel. I did not check the reproducibility of the experiments.

**Strength And Weaknesses:**

Strength:
1. The authors proposed a pruning method that can be applied before training and without training data. The method shows better performance compared with existing methods in some cases.
2. Several tricks are proposed to alleviate the computational cost of the method, such as finite difference and new-input-new-weight trick.

Weaknesses:
As the source of intuition, whether preserving the NTK spectrum is really the reason that NTK-SAP works is not discussed. I tend to be skeptical because
1. The NTK dynamics approximates neural network dynamics only when the network is very large. Usually used networks are not large enough to enter the kernel regime, especially for those after pruning.
2. For practical neural networks, the NTK can change significantly during training, thus keeping the spectrum of initial NTK may not be important.
3. Since the Jacobi is a very large matrix, approximating it using finite difference can introduce large error. Also, error may appear in the weight sampling step.
To address these concerns, I suggest the authors do more investigation on the NTKs and NTK dynamics of the networks before and after pruning. For example, discussion can be made on:
1.Does the NTK spectrum change a lot over training?
2.How close is the approximated spectrum to the real one? How large is the variance?
3.Does the pruned network still have similar NTK spectrum with the dense network after some training?

Of course, even if the NTK cannot explain why the proposed method works, the method can still be useful, and there may be more interesting mechanism behind the performance of the method.

**Summary Of The Paper:**

In this paper the authors proposed a foresight pruning method for the pruning of neural networks by maintaining the spectrum of the neural tangent kernel (NTK) of the network during the pruning. The quantity that the proposed method, named NTK-SAP, tries to keep is the trace of the NTK matrix. To save computational cost, a finite difference is used to approximate the Jacobi matrix of the network. To obtain good estimate of the trace of the analytic NTK, multiple weights are sampled from the initialization distribution to compute an average of fixed-weight-NTK. A "new-input-new-weight" approach is used to lower computational cost. Finally, by replacing the input data by random Gaussian, the method is made data agnostic. Numerical experiments are conducted, showing that the NTK-SAP method has better performance compared with existing methods, especially when the pruning rate is large.

**Summary Of The Review:**

This paper proposes a new method to prune neural networks without training. The method can be made data-agnostic. Experiments in the paper show that the method can provide better pruned model than existing methods, especially when the pruning rate is high. The application part is interesting. It will be more interesting if the real mechanism behind the good performance of the proposed method can be further explored.

---

> ### Author Response · Authors · 2022-11-14
> **Response to reviewer 8ZBV [Part 3/3]**
>
> > Does the pruned network still have similar NTK spectrum with the dense network after some training?
>
> `A6` The short answer is: the pruned network does not necessarily have a similar NTK spectrum with the dense network after some training, but we did not intend to make the eigenspectrum of the pruned network close to the dense counterpart over the **entire** training process. We elaborate below.
>
> To start with, let us clarify the complete motivation and intuition of our work as follows:
>
> The analytic NTK is a good measure of the optimization of the large enough network since it measures the expected behavior over random weight configurations and training trajectory. Let
> $\boldsymbol{\theta}_t(\boldsymbol{\theta}_0;t)$ denotes the parameter of the network at training step $t$ with initial random weight configuration $\boldsymbol{\theta}_0$, then:
>
> $$
> \boldsymbol{\Theta}(\mathcal{X},\mathcal{X})
> =
> \mathbb{E}_{\boldsymbol{\theta}_0, t}
> \left[ \underset{ n_l\rightarrow\infty, \forall l}{\lim}
> \hat{\boldsymbol{\Theta}}^{\\{n_l\\}}(\mathcal{X},\mathcal{X}; \boldsymbol{\theta}_t(\boldsymbol{\theta}_0;t))\right].
> $$
>
> Many existing works compute the fixed-weight-NTK with a single random weight configuration at initialization and are hence not a good measure of the optimization of the network with practical size. In our work, we compute the expected fixed-weight-NTK and we believe such an expected version of the fixed-weight-NTK better captures the expected behavior of the subnetworks over random weight configurations and training trajectory. Hence, we aim to find a subnetwork with a well-conditioned expected fixed-weight-NTK.
>
> To make the expected fixed-weight-NTK well-conditioned, we choose to align it to the fixed-weight-NTK of the dense network at initialization. Such a choice is justified for the following two reasons:
>
> 1. **The eigenspectrum of the dense fixed-weight-NTK is a good candidate.** We believe that the good performance of the dense network implies that the eigenspectrum of the dense network at initialization is well-conditioned (See Section 6.5 and Appendix G for the eigenspectrum of the dense network.).
>
> 2. **The eigenspectrum of the dense fixed-weight-NTK is a practical choice.** For foresight pruning, we can only access the fixed-weight-NTK at initialization. Also, the foresight pruning framework mentioned in Section 3 is a general framework that preserves a scalar indicator of the dense network. Hence, due to the intrinsic nature of the framework, we can only try to align the eigenspectrum of the sparse network to the dense counterpart at initialization, rather than some given well-conditioned spectrum.
>
>
> Indeed, for practical networks, the eigenspectrum of the dense network may be changing during the training procedure. Hence, our method can only approximately align the training behavior to the dense network at the early training stage. For the later epochs, the expected well-conditioned property of the subnetwork ensures that the determined subnetworks have a better optimization performance compared to other foresight pruning methods.
>
> In Appendix G.2, we also empirically compare the eigenvalue distribution of the fixed-weight-NTK for different foresight pruning methods along with the dense counterpart. We see that indeed the spectrum of the dense network is changing throughout the training. On the contrary, the fixed-weight-NTK eigenspectrum of all the pruned networks shows similar behavior and deviates from the dense one. However, we showed that, compared to other foresight pruning methods, the eigenspectrum of NTK-SAP aligned with the dense counterpart for the first few epochs (As it is shown that the majority of the eigenvalues of NTK-SAP are the closest to the dense network; also reflected in Appendix G.1.). Secondly, the fixed-weight-NTK of NTK-SAP is well-conditioned throughout the entire training process, which aligns with our motivation. We have also incorporated the discussion into Appendix B.2.
>
> [1] Chen, Wuyang, Xinyu Gong, and Zhangyang Wang. "Neural Architecture Search on ImageNet in Four GPU Hours: A Theoretically Inspired Perspective." International Conference on Learning Representations. 2020.
>
> [2] Xu, Jingjing, et al. "KNAS: green neural architecture search." International Conference on Machine Learning. PMLR, 2021.

---

> ### Author Response · Authors · 2022-11-14
> **Response to reviewer 8ZBV [Part 2/3]**
>
> > Since the Jacobi is a very large matrix, approximating it using finite difference can introduce large error. Also, error may appear in the weight sampling step.
>
> `A3` Indeed, the finite difference method and weight sampling can introduce errors. To check if our algorithm works, we have indeed included the spectrum of the fixed-weight-NTK of NTK-SAP with the dense baseline in Figure 2 and Appendix G.1. It shows that NTK-SAP indeed preserves the fixed-weight-NTK eigenspectrum of the dense network compared to other foresight pruning methods. In this sense, we believe the finite difference method approximation is useful.
>
> > I suggest the authors do more investigation on the NTKs and NTK dynamics of the networks before and after pruning. For example, discussion can be made on: 1.Does the NTK spectrum change a lot over training?
>
> `A4` In Appendix G.2, we have included the relative condition number of the fixed-weight-NTK of different foresight pruning methods. We find that the relative eigenvalue (hence condition number) of NTK-SAP is almost unchanged during training and more well-conditioned. That may explain the superior performance of NTK-SAP and also reflect the intention of NTK-SAP, which tries to make pruned networks well-conditioned throughout training.
>
> > 2. How close is the approximated spectrum to the real one?
>
> `A5` We are assuming that the reviewer is referring to how accurate our approximation is compared to the analytic NTK. Unfortunately, to the best of our knowledge, we are unable to easily obtain the expression of the analytic NTK for modern networks, including ResNets. However, we have included a few comparison experiments with the fixed-weight-NTK, which can be found in `A3`, `A4`, and `A6`.

---

> ### Author Response · Authors · 2022-11-14
> **Response to reviewer 8ZBV [Part 1/3]**
>
> We thank the reviewer for recognizing our work is novel and well-written. Below please find the responses to your questions and we sincerely hope they could address your concerns regarding NTK theory.
>
> > The NTK dynamics approximates neural network dynamics only when the network is very large. Usually used networks are not large enough to enter the kernel regime, especially for those after pruning.
>
> `A1` We first make some clarifications on the NTK theory our method is based on. As claimed by many theoreticians, the reviewer may think that "NTK theory works for lazy training" or "NTK theory can only be applied to gradient descent near the initial point". Nevertheless, there are many versions of NTK theories. One "common" version claims that gradient descent (GD) converges for ultrawide networks, which is where the claim "NTK theory can only be applied to gradient descent near initial point" comes from (since in this case, the trajectories of GD are indeed confined near initial point). However, there is another "eigenvalue" version: if the eigenvalues of NTK remain positive during training, then GD converges and its convergence rate depends on the eigenvalues of the NTK matrix. In this version, GD is not necessarily confined in the neighborhood of the initial point as long as the condition for the eigenvalues holds. (We would like to add that in the "common" version, the condition on the width or neighborhood regime actually guarantees that the eigenvalues of NTK remain positive, so the "eigenvalue" version is more general than the "common" version).
>
> Consequently, by the description of the "eigenvalue" version, we only need to keep the eigenvalues of the NTK matrix well-conditioned during training. We do not need GD to stick to the NTK (kernel) regime.
>
> > For practical neural networks, the NTK can change significantly during training, thus keeping the spectrum of initial NTK may not be important.
>
> `A2` In the previous question, we have explained the "eigenvalue" version of the NTK theory, where we need to pay attention to the condition number of the possibly time-varying eigenspectrum. In this question, we would like to give more details from two perspectives.
>
> **Is the spectrum of initial NTK not important?** We agree that for practical neural networks, the fixed-weight-NTK (and its spectrum) may be changing throughout training. However, does it mean that the fixed-weight-NTK at initialization is not informative at all? The answer is actually no. In fact, several works [1, 2] in the research field of neural architecture search (NAS) showed empirically that the spectrum of the fixed-weight-NTK at initialization is correlated to the test performance of neural networks. Hence, the spectrum of initial NTK is indeed important.
>
> **Possibly going beyond just the initial NTK.** Furthermore, to possibly account for the eigenspectrum change during training, we want to take another step further by considering the average statistics of the fixed-weight-NTK over multiple random configurations. The motivation stems from two perspectives. Firstly, it is natural to consider the expectation due to the stochastic nature of the analytic NTK computation (See Appendix B.1 for detailed discussion). Secondly, as long as the time-varying fixed-weight-NTK is well-conditioned throughout training, a better convergence of the neural network can be guaranteed (inspired by the "eigenvalue" version of the NTK theory). To this end, computing the average over multiple weight configurations is more likely to capture the expected behavior of the neural network in the parameter space as it tends to make the fixed-weight-NTK well-conditioned with higher probability (See Appendix B.2 for a more detailed discussion). In fact, we indeed found that the relative eigenspectrum for NTK-SAP is almost unchanged during training in Appendix G.2, hence well-conditioned. Please also see our response `A6` for a detailed discussion of the empirical results.

---

### Official Review · Reviewer_w2XF · 2022-10-25

**Confidence:** 4
**Correctness:** 2
**Technical Novelty And Significance:** 4
**Empirical Novelty And Significance:** 3
**Recommendation:** 6

**Clarity, Quality, Novelty And Reproducibility:**

Clarity and quality: as mentioned above, I feel that these could be improved by checking that the claimed equivalence between the proposed saliency score and the motivation using the NTK.

Novelty: the method is AFAIK novel, previous pruning methods using the NTK are appropriately discussed.

Reproducibility: the algorithm is well explained and the used hyperparameters are provided, so I assume that the empirical results are reproducible.

**Strength And Weaknesses:**

It looks to me that you could have written the same paper with no mention of the NTK since eq. 10 suffices by itself to motivate your method: saliencies are computed so that on average, the pruned function is similar to the original one, as measured by the l2 norm, even after training in directions $\Delta \theta$.

If you posit that the reason for your method to give good empirical results is linked to the theory around NTK, I think that the series of steps that you follow to link your saliency scores (eq. 10) to the intuition using the NTK should be carefully checked, either:
 - theoretically by providing closed-form expressions for the variance of the estimator on simpler models
 - empirically on models for which you are able to actually compute NTKs

The fact that your method is weight-agnostic is also in my opinion not explored as much as it should: did you check which parameters are actually pruned, corresponding to what layers? Given the symmetry invariance of deep nets if e.g. you swap neurons A and B of the same layer (and swap accordingly the corresponding parameters of the previous and next layers), why should neuron A be pruned rather than neuron B ? At high sparsity ratio, how do you ensure that not all parameters of a layer are pruned?

**Summary Of The Paper:**

A new pruning method for deep networks is proposed. The method is motivated as computing weight saliencies so that after pruning, the spectrum of the NTK remains as close as possible to the original NTK. In practice, a series of tricks are used to make computing saliencies more efficient. At the end of the day, saliencies are computed using eq. 10 that is claimed to approximately ensure that the spectrum of the NTK is kept close to the original one. The claimed equivalence between eq. 10 and the intuition using the NTK is made through a series of steps, which I tentatively summarize here:

 1. estimate the spectrum of the infinite width NTK using an average of finite-width NTKs
 2. summarize the spectrum of the NTK using the trace norm of the NTK or equivalently the frobenius norm of the weight-output jacobian
3. estimate the trace norm of the jacobian by computing its product with normally distributed parameter space vectors $\Delta \theta$
4. estimate the jacobian using finite difference
5. estimate saliencies of randomly drawn input vectors from a normal distribution
6. NINW trick where all random quantities are re-sampled at every mini-batch in order to improve efficiency

The method is tested on several datasets and architectures, demonstrating improved test accuracy compared to other benchmarks. A sensitivity analysis to hyperparameters is provided.

**Summary Of The Review:**

A sound new method for computing saliency scores for pruning deep nets but I am not entirely buying the link between the method and the proposed intuition using the NTK framework.

---

> ### Author Response · Authors · 2022-11-14
> **Response to reviewer w2XF [Part 2/2]**
>
> > If you posit that the reason for your method to give good empirical results is linked to the theory around NTK, I think that the series of steps that you follow to link your saliency scores (eq. 10) to the intuition using the NTK should be carefully checked, either: theoretically by providing closed-form expressions for the variance of the estimator on simpler models; empirically on models for which you are able to actually compute NTKs.
>
> `A2` We thank the reviewer for the helpful suggestion. Empirically, we have indeed included the spectrum comparison of the fixed-weight-NTK between pruned networks and dense networks in Section 6.5 and Appendix G. The results show that NTK-SAP can indeed preserve the eigenspectrum of the fixed-weight-NTK at initialization. Hence, we have indeed checked that our method preserves the NTK eigenspectrum at initialization, and we believe that the good empirical results are also linked to the well-conditioned fixed-weight-NTK throughout training (Please also see our response to reviewer 8ZBV).
>
> As for the theoretical perspective, to the best of our knowledge, the analytic NTK cannot be easily derived and computed for modern networks like ResNets. We will leave such an exploration to future research.
>
> > The fact that your method is weight-agnostic is also in my opinion not explored as much as it should: did you check which parameters are actually pruned, corresponding to what layers? Given the symmetry invariance of deep nets if e.g. you swap neurons A and B of the same layer (and swap accordingly the corresponding parameters of the previous and next layers), why should neuron A be pruned rather than neuron B?
>
> `A3` We thank the reviewer for raising such an interesting question. Indeed our method is not really able to distinguish the difference between two neurons in the same layer. This is our intention and the intention was inspired by the results from [1]. More specifically, [1] states that most foresight pruning methods are not very sensitive to layer-wise mask shuffling. Hence, most foresight pruning methods are actually finding suitable layer-wise sparsity ratios. To conclude, NTK-SAP is indeed trying to identify good layer-wise sparsity ratios but not different neurons/weights in the same layer.
>
> > At a high sparsity ratio, how do you ensure that not all parameters of a layer are pruned?
>
> `A4` We thank the reviewer for again raising an important question. This problem where all parameters of a layer are entirely pruned is called layer collapse and is discussed in Synflow [2] and Iterative SNIP [3]. Here we answer the questions as follows:
>
> SPP [4] is a re-initialization method based on SNIP aiming at improving signal propagation when layer collapse is about to happen. This work reveals that when layer collapse is about to occur, the spectrum of the layer-wise input-output Jacobian is extremely poor so it breaks layer-wise dynamical isometry. Similarly, the spectrum of the NTK can also serve as such an indicator. Specifically, it can be proved that when layer collapse is about to happen, the spectrum of the NTK deteriorates as well. Here, we would like to give a simple example for a better understanding. For example, when layer collapse happens for layer $\ell$, then the weight gradients of DNNs of all preceding layers will be 0, and NTK-SAP will implicitly prevent this from happening. In this sense, NTK-SAP works in a way very similar to Iterative SNIP [3] to avoid layer collapse. This discussion can also be found in Appendix N.
>
> [1] Frankle, Jonathan, et al. "Pruning Neural Networks at Initialization: Why Are We Missing the Mark?."International Conference on Learning Representations. 2020.
>
> [2] Tanaka, Hidenori, et al. "Pruning neural networks without any data by iteratively conserving synaptic flow."Advances in Neural Information Processing Systems 33 (2020): 6377-6389.
>
> [3] de Jorge, Pau, et al. "Progressive Skeletonization: Trimming more fat from a network at initialization."International Conference on Learning Representations. 2020.
>
> [4] Lee, Namhoon, et al. "A Signal Propagation Perspective for Pruning Neural Networks at Initialization."International Conference on Learning Representations. 2019.

---

> > ### Comment · Reviewer_w2XF · 2022-11-18
> > **Thanks for your response**
> >
> > I don't think that your response answers my remark that eq. 10 suffices by itself, even if you are indeed right that I could have been more precise in putting it into words.
> >
> > Regarding figure 2 and other figures in appendix, I am a bit skeptical about the use of the log-scale on the y-axis. It is well known that the spectrum of the NTK is concentrated on a few number of large eigenvalues, whereas using a log-scale emphasizes small values. On figure 2 e.g. it looks like for high eigenvalues, iterative SNIP is closer to the baseline. In addition, by only comparing eigenvalues, you might miss different eigenvectors. I suggest you perhaps use a more systematic measures such as kernel alignment [1].
> >
> > In general, I suggest you conduct empirical evaluation of all approximation steps (1-6 in my initial review). This is to confirm that the reason for your method to work is actually related to how you motivate it.
> >
> > [1] Nello Cristianini, John Shawe-Taylor, André Elisseeff and Jaz S. Kandola. "On kernel-target alignment"

---

> > > ### Author Response · Authors · 2022-11-29
> > > **Further response to the reviewer**
> > >
> > > We thank the reviewer for the comments. We would like to answer the reviewer's feedback as follows:
> > >
> > > > I don't think that your response answers my remark that eq. 10 suffices by itself, even if you are indeed right that I could have been more precise in putting it into words.
> > >
> > > We thank the reviewer for the response, though it is somewhat short, and vague to us with incomplete logical steps. We tried our best to understand how the reviewer thinks but there still exist some difficulties. The reviewer thinks "the pruned function is similar to the original one" (we will use "output preservation" for brevity) motivates a salience score ("saliences are computed so that on average..."), which we call "salience score A". As a result, "output preservation" motivates "our method" through "Eqn. 10". However, we believe the logic chain is flawed.
> > >
> > > We try to complete the reviewer's logic chain by discussing the relationship between the two mentioned salience scores, i.e. "salience score A" and "Eqn. 10". Two potential possibilities can make the logic chain work: (1) "salience score A" is exactly "Eqn. 10"; (2) "salience score A" is similar to "Eqn. 10" in a way that they both have function outputs in the expressions. However, both completed versions of the logic chains still do not recognize the difference between the two from multiple perspectives, as explained below.
> > >
> > > The first possibility neglects the difference in the expression level. We would like to kindly point out that we have made a discussion about the difference between the exact expressions of "salience score A" and "Eqn. 10".
> > >
> > > The second possibility is flawed because it neglects the difference in the meaning level. The reviewer is making a hand-wavy claim without important steps which leads to the neglect. Once again, though "salience score A" and "Eqn.10" have outputs in their expressions, they represent different meanings, i.e. function output difference, and weight-output Jacobian difference. NTK, as the gram matrix of the weight-output Jacobian, is a natural intuition for "Eqn. 10" while "output preservation" is not. If the reviewer thinks it is, we will be glad to hear about more details on how to derive Eqn. 10 from "output preservation", since we could not figure out how.
> > >
> > > > Regarding figure 2 and other figures in appendix, I am a bit skeptical about the use of the log-scale on the y-axis. It is well known that the spectrum of the NTK is concentrated on a few number of large eigenvalues, whereas using a log-scale emphasizes small values. On figure 2 e.g. it looks like for high eigenvalues, iterative SNIP is closer to the baseline.
> > >
> > > We thank the reviewer for the comments. We justify why we present the eigenvalues in a log scale to emphasize small (and other) eigenvalues in the log scale:
> > >
> > > 1. Small eigenvalues are more important than the largest eigenvalues in terms of neural network optimization, as most existing works [1,2,3] show that the smallest eigenvalue can be used to upper-bound the training losses.
> > >
> > > 2. All eigenvalues are important as they control the convergence of different eigen-modes.
> > >
> > >
> > > > In addition, by only comparing eigenvalues, you might miss different eigenvectors.
> > >
> > > As mentioned in our paper, we are only interested in aligning the NTK spectrum rather than the NTK matrix itself. Hence, we do not pay much attention to NTK eigenvectors.
> > >
> > > > I suggest you perhaps use a more systematic measures such as kernel alignment. In general, I suggest you conduct empirical evaluation of all approximation steps (1-6 in my initial review). This is to confirm that the reason for your method to work is actually related to how you motivate it.
> > >
> > > We want to first kindly point out that the step 1 in the suggestion provided by the reviewer is impractical because:
> > >
> > > 1. It is not yet easy to derive the formulation of the analytic NTK for arbitrary modern networks.
> > >
> > > 2. Deriving the analytic NTK may be a good topic for a theory paper, but definitely goes much beyond the scope of our work (a more empirical work) and should be left for future research.
> > >
> > >
> > > And we have indeed (more or less) shown the comparison from steps 2 to 6 via the eigenspectrum of the fixed-weight-NTKs.
> > >
> > > [1] Xu, Jingjing, et al. "KNAS: green neural architecture search."*International Conference on Machine Learning*. PMLR, 2021.
> > >
> > > [2] Lee, Jaehoon, et al. "Wide neural networks of any depth evolve as linear models under gradient descent."*Advances in neural information processing systems*32 (2019).
> > >
> > > [3] Xiao, Lechao, Jeffrey Pennington, and Samuel Schoenholz. "Disentangling trainability and generalization in deep neural networks."*International Conference on Machine Learning*. PMLR, 2020.

---

> ### Author Response · Authors · 2022-11-14
> **Response to reviewer w2XF [Part 1/2]**
>
> We thank reviewer w2XF for recognizing our work is novel and our algorithm is well-explained. Below please find the responses to your concerns and we hope they could address your concerns:
>
> > It looks to me that you could have written the same paper with no mention of the NTK since eq. 10 suffices by itself to motivate your method: saliencies are computed so that on average, the pruned function is similar to the original one, as measured by the $\ell_2$ norm, even after training in directions $\Delta \theta$.
>
> `A1` We thank the reviewer for initiating an interesting discussion about the motivation of our work. However, we would like to kindly point out that the interesting motivation provided by the reviewer does not directly lead to Equation 10 in our work. We will formulate the saliency score corresponding to the reviewer’s motivation.
>
> | Space\Time        | At initialization | Move in direction $\Delta \theta$ |
> |:-----------------:|:-----------------:|:---------------------------------:|
> | Original function | $f_1$             | $f_2$                             |
> | Pruned function   | $f_3$             | $f_4$                             |
>
> Firstly, we will start with the formulation so that "the pruned function is similar to the original one" as the reviewer suggested. We have also included a table above to have a better visualization of which functions we are referring to. Denote the original function as $f_1=f(x;\theta_0^j \odot m^j)$, where $\theta_0^j$ is a specific individual weight and $m^j=1$ is the corresponding mask. Then, "the pruned function" is $f_3=f(x;\theta_0^j \odot (m^j-1))$. To make them similar "measured by the $\ell_2$ norm", the algorithm would control $S_0(m^j) = \Delta f^j=\\|f_1-f_3\\| \_2=\\|f(x;\theta_0^j \odot m^j)-f(x;\theta_0^j \odot (m^j-1))\\| \_2$. As mentioned in the original paper of SNIP, computing the saliency score $S_0$ for each parameter is prohibitively expensive since it requires $p+1$ number of forward passes, where $p$ is the number of parameters. Hence, the following approximation $S_1$ is used: $\Delta f^j \approx S_1(m^j)= \lim_{\delta \rightarrow 0} 1/\delta \\|f(x;\theta_0^j \odot (m^j-\delta))-f(x;\theta_0^j \odot m^j)\\| \_2= \left. \partial \\|f(x;\theta_0^j \odot m^j)\\| \_2/\partial m^j \right|_{m^j=1}=\partial \\|f_1\\| \_2/\partial m^j$. Hence, to preserve the output of the neural network measured by the $\ell_2$ norm, the saliency score we should use is $S_1(m^j)=|\partial \\|f(x;\theta_0^j \odot m^j)\\| _2/\partial m^j|=|\partial \\|f_1\\|_2/\partial m^j|$ and weights with the smallest $S_1$ will be pruned.
>
> The reviewer mentioned that we should consider functions "even after training in directions $\Delta \theta$". We think the reviewer refers to replacing $\theta_0$ in $S_1$ by $\theta_0+\Delta \theta$, leading to $S_2(m^j)=|\partial \\|f(x;(\theta_0^j+\Delta \theta) \odot m^j)\\|_2/\partial m^j|=|\partial \\|f_2\\|_2/\partial m^j|$ (here we neglect the expectation over $\theta_0$ and $\Delta \theta$ for simplicity).
>
> On the contrary, notice that the saliency score of our formulation, i.e. Equation 7 (or Equation 10), is different from $S_1$ and $S_2$, i.e., $S_3 = |\partial \\|f(x; (\theta_0^j+\Delta \theta) \odot m^j) - f(x; \theta_0^j \odot m^j)\\|_2/\partial m^j| \approx | \\|f_2-f_1\\|_2-\\|f_3-f_4\\|_2|$. Notice that our saliency score $S_3$ has a very different expression compared to $S_1$ and $S_2$. That is because intuitively, the saliency score $S_3$ measures **the change of the parameter-output Jacobian (NOT the change of function output in $S_1$ or $S_2$)** before and after pruning.
>
> Put it simply, the reviewer wants to control $\\|f_2-f_4\\|_2$ (and maybe potentially $\\|f_1-f_3\\|_2$) while we are trying to control $\\|f_2-f_1\\|_2-\\|f_3-f_4\\|_2$. These two terms are non-trivial and not obvious without the help of the NTK theory.

---

### Official Review · Reviewer_B3ZS · 2022-10-25

**Confidence:** 3
**Clarity, Quality, Novelty And Reproducibility:** 1. The paper is clearly written.
2. …
**Correctness:** 3
**Technical Novelty And Significance:** 3
**Empirical Novelty And Significance:** 2
**Recommendation:** 6

**Details Of Ethics Concerns:**

No ethics concerns.

**Strength And Weaknesses:**

Strength:

1. Foresight pruning is an important research direction, as well as the explorations on weight-agnostic and data-agnostic pruning.
2. NTK-SAP is a simple and novel pruning metric. Choosing the average value of all eigenvalues of NTK as saliency metric should be able to capture some valid spectrum information while providing computational benefits.
3. The experiments and analysis show NTK-SAP can preserve the NTK spectrum and leads to better accuracy.

Weakness:

1. Since the approximation requires two forward passes, the method still induces larger computational cost.

**Summary Of The Paper:**

This paper proposed a new foresight pruning metric, which prunes the connections that have the least influence on the trace norm of the NTK. The motivation is to align the training dynamics of the sparse and dense model by maintaining the spectrum of NTK. To alleviate the burden of estimating the expectation of the fixed-weight-NTKs, the authors propose to sample a weight configuration for each mini-batch and compute the averages of their fixed-weight-NTKs.

**Summary Of The Review:**

NTK-SAP explores an important pruning topic, foresight pruning. The method is novel by considering preserving the spectrum of NTK. The experiment results demonstrate its effectiveness over existing baselines, though at the expense of some computational efficiency.

---

> ### Author Response · Authors · 2022-11-14
> **Response to reviewer B3ZS**
>
> We thank reviewer B3ZS for recognizing our work is well-motivated, simple, novel, and clearly-written with comprehensive experiments. We hereby answer the reviewer’s question as follows:
>
> > Since the approximation requires two forward passes, the method still induces a larger computational cost.
>
> `A1` We thank the reviewer for pointing out such an important question regarding the computational cost of NTK-SAP. In fact, we totally agree with reviewer B3ZS that computational cost is crucial in making an algorithm practical. We have indeed included such a discussion in the last paragraph of Section 5.3, Section 6.4, and Appendix D. We have also included the detailed computational time on NVIDIA V100 GPUs here:
>
> |                  | CIFAR-10 | CIFAR-100  | Tiny-ImageNet | ImageNet(ResNet18) | ImageNet(ResNet50) |
> | ---------------- |:--------:|:----------:|:-------------:|:------------------:|:------------------:|
> | 1 epoch training | 9.6 s    | 11 s       | 24 s          | 15 min 42 s        | 18 min             |
> | SNIP             | 1 s      | 1 s        | 1 s           | 13 s               | 34 s               |
> | GraSP            | 1 s      | 1 s        | 3 s           | 50 s               | 5 min 32 s         |
> | Synflow          | 12 s     | 34 s       | 1 min 2 s     | 6 min 34 s         | 5 min 39 s         |
> | Iterative SNIP   | 24 s     | 46 s       | 1 min 18 s    | 16 min 42 s        | 23 min 17 s        |
> | NTK-SAP          | 17 s     | 1 min 14 s | 2 min 14 s    | 17 min 53 s        | 22 min 49 s        |
> | NTK-SAP-S        | 3 s      | 14 s       | 31 s          | -                  | -                  |
>
> We would like to further explain here why such extra computation cost is reasonable:
>
> **NTK-SAP does not need knowledge of the dataset.** Synflow and NTK-SAP, unlike data-dependent foresight pruning methods (GraSP and SNIP), only need to know the resolution of the images in the dataset. That is to say, found masks can be directly used for different datasets as long as they have the same resolution (or resized to have the same resolution). Such a property circumvents the computational cost of identifying unique masks for different datasets, which is more advantageous when a large dataset is considered, e.g., ImageNet.
>
> **The computational cost of the NTK-SAP is not significant considering the entire training procedure.** One advantage of foresight pruning is that the pruning procedure only happens once before the training starts. As is shown in the table above (also can be found in Appendix D Table.3), the pruning cost of NTK-SAP (or NTK-SAP-S, which has a smaller pruning round T) is usually lower than 1-2 epochs of neural network training. Hence, the computational cost only contributes to less than 1%-2% of the total training cost, considering commonly adopted 100-200 training epochs. Consequently, such increased pruning cost is reasonable. Moreover, NTK-SAP can sometimes find networks with less inference cost compared to other foresight pruning methods, which further reduces the training cost of determined sparse models. Please refer to Section Appendix D.2 for more details.

---

### Official Review · Reviewer_2M9f · 2022-10-31

**Confidence:** 3
**Correctness:** 3
**Technical Novelty And Significance:** 3
**Empirical Novelty And Significance:** 3
**Recommendation:** 6

**Clarity, Quality, Novelty And Reproducibility:**

**Clarity**: The paper is well-written and easy to follow.
**Quality**: The approach combines ideas from analyzing training dynamics and NN-pruning to suggest NTK-SAP outperforms strong baselines across datasets.
**Novelty**: The algorithm is novel, to my knowledge.
**Reproducibility**: The authors provide sufficient information for reproducing critical results from the paper.

**Strength And Weaknesses:**

### **Strengths**
+ The problem is well-motivated; pruning neural networks by aligning the dynamics could be a good diagnostic tool for dynamics-aware pruning (and potentially improve the generalization of the sparse models).
+ Authors provide exhaustive comparisons to strong baselines, outperforming most approaches across multiple datasets.


### **Weaknesses**
- The iterative pruning algorithm poses significant computational overhead as compared to baselines. Therefore, comparing the runtime/time complexity of NTK-SAP with baselines would help evaluate the approaches head-to-head.
- The plots don't present a linear x-axis for the sparsity ratios. Is it logarithmic along the x-axis? Discussion on the choice of the evaluation checkpoints and quality of performance degradation with a reduction in parameters would be helpful.

**Summary Of The Paper:**

The authors consider the problem of improving the training efficiency of neural networks by pruning them at initialization. Unlike prior literature work, which often considers weight-dependent metrics to find "dead units" and remove them, the authors provide a spectrum-aware metric for identifying removable units. In particular, the critical insight of this work relies on how well the NTK approximates the training dynamics of neural networks beyond the infinitely-wide setting. The authors show that by removing connections that least impact the NTK spectrum during training, they achieve competitive (and often better) performance when compared to solid baselines (e.g., SNIP). To provide a computationally efficient pruning algorithm, the authors identify some challenges in estimating the NTK spectrum at convergence and show that when averaged appropriately, multiple random initializations of the NN provide a good approximation for the spectrum.

**Summary Of The Review:**

The authors consider alternative metrics for searching connections for pruning while maintaining training performance and improving learning efficiency. Unlike weight-dependent metrics, the authors consider the NTK-spectrum as a proxy for evaluating the 'goodness' of connections and propose removing connections that don't perturb the NTK-spectrum post-pruning. Finally, the authors assess some computational bottlenecks in the efficient computation of the spectrum and offer well-motivated approximations that significantly reduce the overhead. With extensive experiments on CIFAR10/100, Tiny-Imagenet, and across model families (ResNet/VGG), NTK-SAP performs quite well compared to alternative approaches in the literature.

---

> ### Author Response · Authors · 2022-11-14
> **Response to reviewer 2M9f [Part 2/2]**
>
> > The plots don't present a linear x-axis for the sparsity ratios. Is it logarithmic along the x-axis? Discussion on the choice of the evaluation checkpoints and quality of performance degradation with a reduction in parameters would be helpful.
>
> `A2` Thank you for the question and helpful suggestion. Actually, the x-axis for Fig.1 is in the logarithmic scale of density (density=1-sparsity). More specifically, we obtain each sparsity ratio by removing 20% of the remaining parameters twice. For example, the first sparsity ratio is 1-0.8*0.8=0.36. We choose these sparsity ratios to evaluate mainly following the conventional practice used in [1]. The intuition is that for the first few rounds, the network is over-parameterized and a large portion of the weights can be removed. This discussion is also added to Appendix A.4 with highlighted color.
>
> We have also added a paragraph discussing the performance degradation with a reduction in parameters in Section 6.1 with highlighted color. We would like to also add it here:
>
> Generally, the test performance of all foresight pruning methods degrades with more sparsity induced, and the speed of performance degradation accelerates as the sparsity ratio increases. Additionally, for VGG-16 (CIFAR-100) and ResNet-18 (Tiny-ImageNet), it can be clearly seen that the test performance of NTK-SAP first increases and then decreases, which shows the generalization improvements brought by pruning. However, it is not observed in ResNet-20 (CIFAR-10).
>
> [1] Frankle, Jonathan, et al. "Pruning Neural Networks at Initialization: Why Are We Missing the Mark?."International Conference on Learning Representations. 2020.

---

> ### Author Response · Authors · 2022-11-14
> **Response to reviewer 2M9f [Part 1/2]**
>
> We sincerely thank the reviewer for acknowledging that our work is well-motivated and our experiments are extensive. Below please find the responses to the reviewer’s concerns:
>
> > The iterative pruning algorithm poses significant computational overhead as compared to baselines. Therefore, comparing the runtime/time complexity of NTK-SAP with baselines would help evaluate the approaches head-to-head.
>
> `A1` We thank the reviewer for the constructive suggestion. We want to kindly point out that we have indeed included such a comparison in our manuscript. Specifically, we included such details in the last paragraph of Section 5.3, Section 6.4, and Appendix D. More precisely, detailed computational time on NVIDIA V100 GPUs and FLOPs of different pruning methods can be found in Appendix D.1 and D.2, respectively. We have also included the detailed computational time of each pruning method below:
> |                  | CIFAR-10 | CIFAR-100  | Tiny-ImageNet | ImageNet(ResNet18) | ImageNet(ResNet50) |
> | ---------------- |:--------:|:----------:|:-------------:|:------------------:|:------------------:|
> | 1 epoch training | 9.6 s    | 11 s       | 24 s          | 15 min 42 s        | 18 min             |
> | SNIP             | 1 s      | 1 s        | 1 s           | 13 s               | 34 s               |
> | GraSP            | 1 s      | 1 s        | 3 s           | 50 s               | 5 min 32 s         |
> | Synflow          | 12 s     | 34 s       | 1 min 2 s     | 6 min 34 s         | 5 min 39 s         |
> | Iterative SNIP   | 24 s     | 46 s       | 1 min 18 s    | 16 min 42 s        | 23 min 17 s        |
> | NTK-SAP          | 17 s     | 1 min 14 s | 2 min 14 s    | 17 min 53 s        | 22 min 49 s        |
> | NTK-SAP-S        | 3 s      | 14 s       | 31 s          | -                  | -                  |
>
> The result shows that NTK-SAP does not introduce much extra computational cost compared to other iterative pruning methods, including Iterative SNIP and Synflow. Indeed, iterative pruning methods generally need more time compared to single-shot pruning methods, including SNIP and GraSP, but we believe it does not prevent NTK-SAP from being practical in real-world application scenarios. We conclude the reasons as follows:
>
> **NTK-SAP is data-agnostic.** Iterative pruning methods including Synflow and NTK-SAP do not require any training data. Hence, once the resolution of the target images is determined, we can pre-compute the masks, which can be directly transferred to any datasets with the same resolution (or resized to the same resolution). On the contrary, SNIP and GraSP, which are data-dependent, need to determine the masks every time they are applied to a new dataset.
>
> **The pruning cost of NTK-SAP can be further reduced with a smaller pruning round T.** We have included an ablation study of NTK-SAP with smaller pruning rounds T in Appendix C. The results show that the performance of NTK-SAP is not greatly affected even with only 1/5 of the original pruning rounds (We call such a variant NTK-SAP-S). Hence, we can reduce the computational cost of NTK-SAP with even smaller pruning rounds.
>
> **The computational cost of the pruning procedure is only a very small fraction of the total neural network training procedure.** As is shown in the table above (It can also be found in Appendix D Table.3 in our submission), the pruning cost of most considered foresight pruning methods is lower than 1-2 epochs of neural network training. That is to say, the pruning cost is usually less than 1% of the total training cost, considering commonly adopted 100-200 training epochs. Hence, such increased pruning cost does not make iterative methods less favored compared to single-shot pruning methods. Moreover, NTK-SAP can sometimes find networks with less inference cost compared to other foresight pruning methods, which further reduces the training cost of sparse models. Please refer to Section Appendix D.2 for more details.

---

### Decision · Program_Chairs · 2023-01-20

**Decision:**

Accept: poster

**Justification For Why Not Higher Score:**

Explained in part 1.

**Justification For Why Not Lower Score:**

Explained in part 1.

**Metareview: Summary, Strengths And Weaknesses:**

The paper proposes a network pruning method at initialization by pruning the connections that have the least influence on the spectrum of the NTK. The goal is to maintain the NTK spectrum, which may help align the training dynamics to that of its dense counterpart. The paper is discussed extensively in a meeting with the reviewers. While the reviewers found the proposed method interesting and effective, there were some concerns on the connection between the theoretical motivation of the paper and the proposed weight-agnostic score used in the experiments. In particular, since several approximations are made to get the final weight-agnostic score, it was not clear if the approximations are accurate and how much error each step makes. I asked the authors to verify the steps empirically on a small (potentially 1- or a few-layer) network to validate the approximations, in particular from Eq 9 to Eq 10 to close this gap and make the paper more convincing. The provided experiments addressed the reviewers concerns, and hence I recommend acceptance. The authors are strongly encouraged to include the new experiment to the main manuscript.

**Note From Pc:**

if the above contains the word "oral" or "spotlight" please see: "oral" presentation means -> notable-top-5% and "spotlight" means -> notable-top-25%. As stated in our emails, we are disassociating presentation type from AC recommendations